# Interpretable Next-token Prediction via the Generalized Induction Head

**Eunji Kim**[1,2*]    **Sriya Mantena**[1,3*]

**Weiwei Yang**[1]    **Chandan Singh**[1]    **Sungroh Yoon**[2,4†]    **Jianfeng Gao**[1†]

[1] Microsoft Research
[2] Department of Electrical and Computer Engineering, Seoul National University
[3] Stanford University
[4] Interdisciplinary Program in Artificial Intelligence, Seoul National University

## Abstract

While large transformer models excel in predictive performance, their lack of interpretability restricts their usefulness in high-stakes domains. To remedy this, we propose the Generalized Induction-Head Model (GIM), an interpretable model for next-token prediction inspired by the observation of "induction heads" in LLMs. GIM is a retrieval-based module that identifies similar sequences in the input context by combining exact n-gram matching and fuzzy matching based on a neural similarity metric. We evaluate GIM in two settings: language modeling and fMRI response prediction. In language modeling, GIM improves next-token prediction by up to 25%p over interpretable baselines, significantly narrowing the gap with black-box LLMs. In an fMRI setting, GIM improves neural response prediction by 20% and offers insight into the language selectivity of the brain. GIM represents a significant step toward uniting interpretability and performance across domains. The code is available at https://github.com/ejkim47/generalized-induction-head.

## 1 Introduction

While modern transformer models have achieved impressive performance across a wide array of next-token prediction tasks [1–3], these models remain black-boxes, limiting their use in real-world applications. Their opacity is detrimental in fields such as neuroscience [4] and social science [5], where trustworthy interpretation, specifically, token-level attribution that traces outputs back to input data, is often the end goal. Their lack of transparency also hinders adoption in high-stakes applications such as medicine [6], raising concerns around regulatory compliance, safety, and alignment [7–10].

As an alternative to these black-box models, interpretable models have been proposed for various tasks [11–13], but they continue to struggle on the task of next-token prediction. For example, in next-token prediction for natural language, the state-of-the-art interpretable model is Infini-gram [14], which trails GPT-2 by 30%p on the BabyLM dataset (see Table 1). Our analysis suggests that this performance gap stems from Infini-gram's inability to adapt to novel contexts or handle minor input variations such as typos and rephrasings.

---

[*]Equal contribution. Work conducted during an internship at Microsoft Research.
[†]Corresponding authors.

39th Conference on Neural Information Processing Systems (NeurIPS 2025).

We address this gap by proposing the Generalized Induction-Head Model (GIM). GIM is inspired by the observation of "induction heads" in LLMs [15, 16] that support in-context learning by detecting and extending patterns in prior input. In pre-trained LLMs, this behavior arises implicitly and is inferred through post-hoc approximations over dense internal states. In contrast, GIM is not a post-hoc tool for interpreting opaque systems, but an inherently interpretable system that explicitly models this behavior in a transparent and auditable manner.

GIM is an interpretable retrieval-based framework that operates entirely within the model's input context to retrieve suggestions for next-token completion. It extends traditional exact matching by incorporating a lightweight fuzzy similarity function to match sequences that yield similar next-token distributions. Importantly, this neural component is used solely for scoring similarity between input phrases rather than generating outputs. The final next-token prediction is computed as a similarity-weighted distribution over tokens that follow the matched phrases, enabling each prediction to be directly attributed to specific input sequences, supporting full interpretability and human auditability. GIM is designed as a standalone, model-agnostic module that supports both exact and fuzzy matching and can be integrated across modalities.

We first evaluate the performance and interpretability of GIM in next-token prediction for language modeling. We integrate GIM into Infini-gram, and GIM improves next-token prediction accuracy by 25%p over Infini-gram using OpenWebText [17] as a reference corpus, significantly narrowing the performance gap with GPT-2 (see Table 1).

Second, we focus on a single, real-world neuroscience problem, deviating from a typical machine-learning conference paper. Grounding in a neuroscience context allows us to avoid common pitfalls in evaluating interpretation methods [18, 19] that seek to test "interpretability" in the abstract. We find that when used to predict fMRI responses to language stimuli, GIM yields a 20% improvement over the state-of-the-art interpretable model (see Table 2), and its transparency enables the attribution of predicted neural responses in each region across the cortex to specific linguistic features.

Taken together, these results challenge the assumption that interpretability and predictive performance are fundamentally at odds, showing that reverse-engineered neural components can be leveraged to enhance transparency. Importantly, our aim is not to claim parity with black-box LLMs, but to show that meaningful gains in predictive performance can be achieved without compromising transparency.

## 2  Related Work

**N-gram language models**    Early language modeling techniques revolved around n-gram models [20, 21], which generally stored next-token probabilities in large tables learned from data [22]. While largely surpassed by neural LLMs, recent works have continued to improve n-gram LMs, e.g., by scaling up the n-gram reference data [23] and improving the n-gram probability representations using suffix arrays and suffix trees [24–26]. This line of work culminated in Infini-gram [14], which efficiently scales n-gram models to massive datasets and is the starting point for our work.

**Bridging interpretable models and LLMs**    Some works have studied bridging n-gram models and LLMs. For example, Khandelwal et al. [27] interpolated neural LMs with an n-gram model and Li et al. [28] trained a neural model to complement an n-gram model. Other approaches augment black-box LMs with nonparametric components, such as $k$-nearest neighbors [27, 29]. While these methods improve performance, they lack transparency in prediction behavior and token-level attribution.

Interpretable models have been proposed for simplified settings such as text classification. Some offer fully interpretable decision processes [30–32], while others offer partial interpretability by approximating model behavior with natural language concepts [33–37]. However, these approaches are not designed for open-ended generation.

In parallel, there has been a recent surge of interest in mechanistic interpretability, which seeks to understand what mechanisms are learned by transformer-based LLMs [38–41]. This line of work identified induction heads in toy LLM models [15] as well as large-scale pre-trained LLMs [42, 16]. Despite these efforts, frameworks to make these findings useful in real settings remain underexplored.

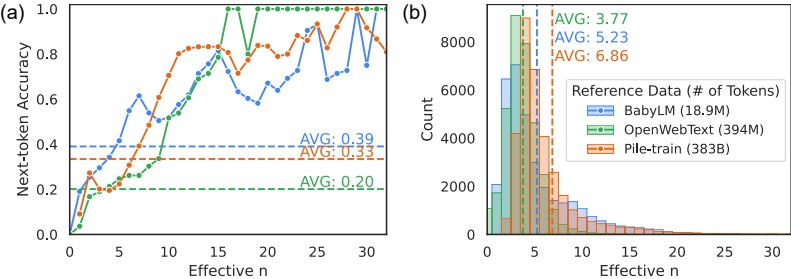

Figure 1: Performance on the BabyLM dataset with Infini-gram built from various reference datasets. (a) Next-token prediction accuracy by effective $n$, with the dashed line indicating the average. (b) The histogram of the count for each effective $n$.

**Natural language representations in fMRI**    In recent years, predicting brain responses to natural language using LLM representations has become common in the field of language neuroscience [43–47]. Predictive "encoding models" have been used to explore the relative contributions of syntax, semantics, and discourse to neural activity [48–55] and to study the cortical organization of language timescales [56, 57], sometimes making use of LLMs to help annotate and generate stimuli [58–61]. However, these models largely operate as black boxes. While they reveal which language representations best predict neural activity, they offer limited insight into where in the cortex these features exert their influence or when in the linguistic stimulus they become relevant.

Separately, behavioral studies have examined how humans recall and process repeated text [62–65] and how similar recall patterns emerge in LLMs [66, 67]. Yet the cortical mechanisms involved in contextual recall remain unclear, motivating our investigation through interpretable modeling.

## 3    Methods

We begin by discussing Infini-gram, the scalable n-gram method for interpretable next-token prediction (Sec. 3.1), and introduce the Generalized Induction-Head Model (GIM) (Sec. 3.2). In describing the method, we focus on the familiar scenario of next-token prediction for language modeling, but note that the method straightforwardly generalizes to generic next-token prediction tasks (e.g., fMRI responses, time series, video frames). We later show how GIM improves interpretable next-token prediction by integrating it with Infini-gram for language modeling (Sec. 4) and combining it with linear regression for fMRI response prediction (Sec. 5).

### 3.1    Preliminaries and Motivation: Infini-gram

Given an input text sequence, Infini-gram [14] searches a reference corpus for the longest exact suffix match to the input, then calculates the next-token distribution based on the token following each of the matches. This search is made efficient by building large-scale suffix arrays that can scale to trillions of reference tokens. The length of the longest match is referred to as the *effective n*, with the accuracy of the estimated probabilities increasing as the *effective n* becomes larger.

Infini-gram is limited by its reliance on exact matches, which becomes problematic under distribution shifts between the input and reference corpus. For instance, when evaluating on the BabyLM[3] [68] test set, Infini-gram built on larger corpora, such as OpenWebText [17], shows lower performance and, on average, has fewer instances of higher effective $n$ compared to the model built on the BabyLM (Fig. 1). With far larger corpora like Pile-train [69], Infini-gram is able to increase the number of instances with a high effective $n$, resulting in improved performance. However, Infini-gram built on BabyLM, which contains only 0.005% of the tokens found in Pile-train, still achieves the highest performance. This highlights the difficulty Infini-gram faces when there is a substantial gap between the reference corpus and the input prompt, making it hard to find matching cases with a large effective $n$. We address this limitation with the concept of the induction head.

---

[3] https://babylm.github.io/

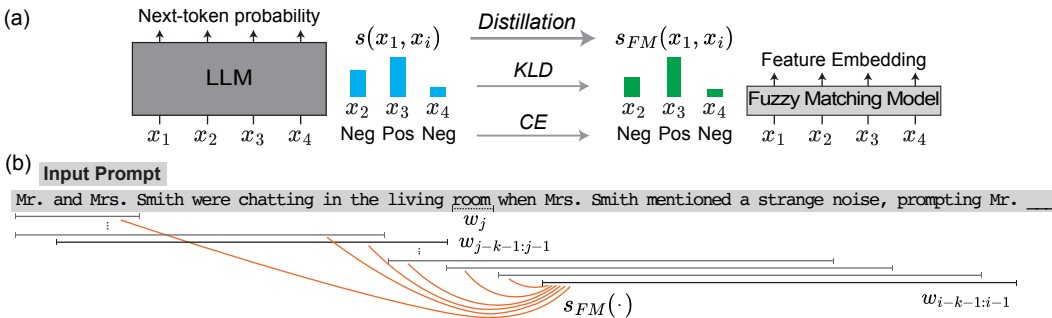

Figure 2: (a) Overview of training Fuzzy Matching Model via distillation from a pretrained LLM. (b) Calculation of sequence similarity within the input prompt for next-token prediction.

## 3.2 Building a Generalized Induction Head

LLMs excel at in-context learning by capturing the statistical distribution of tokens in a given context. One key mechanism enabling this capability is the induction head, a critical component in LLMs responsible for recognizing and extending repeated sequences [15, 16, 42]. Induction heads operate by detecting prior occurrences of a token sequence and leveraging this recurrence for next-token prediction (e.g., given [A][B] ... [A], the model predicts [B]). However, these heads emerge implicitly within LLMs, making their operation difficult to interpret and control.

To this end, we introduce the Generalized Induction-Head Model (GIM) that explicitly models this behavior in a structured, interpretable manner. It functions like Infini-gram but is restricted to the input context. GIM treats the end of the context as a query, searches for the best match within the context and takes the token following the match as the next-token prediction.

**What constitutes a "good match"?** When identifying n-gram-level matches in context, exact matching can perform well if a high effective $n$ is guaranteed (Sec. 3.1), but it can be overly restrictive to minor changes such as rephrasings or typos. To remedy this, we allow for fuzzy n-gram matching, which makes the model more robust to minor changes. Since the fuzzy matching is performed at the level of n-grams, predictions remain interpretable and auditable by a human.

Fuzzy matching requires appropriately computing the similarity between sequences. While similarity can be defined in many ways, in building an induction head, we desire two sequences to be similar if they yield similar next-token distributions. To quantify this, we define the similarity between two sequences, $x_1$ and $x_2$, for fuzzy matching using Jensen–Shannon divergence (JSD):

$$s(x_1, x_2) = \exp\left(-\text{JSD}\left(P_{\text{next}}(x_1), P_{\text{next}}(x_2)\right)\right), \tag{1}$$

where $P_{\text{next}}(\cdot)$ is the estimated next-token probability distribution for a given sequence.

**Computing $s$ efficiently** One approach for computing $s$ would be to use a pre-trained LLM to obtain $P_{\text{next}}$, but this can be computationally expensive. Instead, we develop a small Fuzzy Matching Model, which consists of a few transformer layers and is trained via knowledge distillation from existing LLMs. This model is designed to output feature embeddings that facilitate the calculation of next token probabilities for similarity assessments. With the Fuzzy Matching Model, the similarity between $x_1$ and $x_2$, whose feature embeddings from the model are $e_1$ and $e_2$, is obtained as follows:

$$s_{\text{FM}}(x_1, x_2) = \exp\left(-\left(1 - \text{CosSim}\left(e_1, e_2\right)\right)/T\right), \tag{2}$$

where $T$ is a temperature, which is set to 0.1. The Fuzzy Matching Model is trained with a combination of Cross Entropy (CE) loss and reverse Kullback-Leibler divergence (KLD) loss (Fig. 2(a)). In each training batch, we generate similarity pairs from randomly sampled sequences. The CE loss aids in identifying the most similar pairs. The reverse KLD loss guides the model to follow the overall similarity distribution, ensuring that close pairs receive high scores while distant pairs receive low scores. Further details can be found in Appendix A.2.

**Predicting the next token** Given the similarity scoring function $s_{\text{FM}}$, we construct an induction head that yields the predicted next-token probability distribution $P_{\text{induction}}^{(\text{fuzzy})}$ given an input sequence

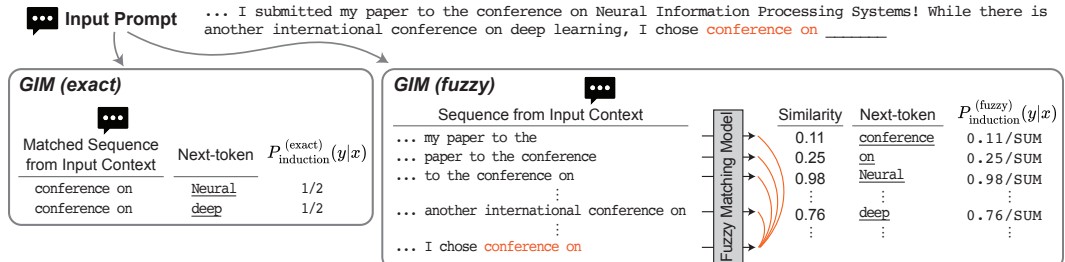

Figure 3: Overview of the GIM pipeline. GIM predicts the next token by efficiently searching for potential next-token completions in the input context with either exact or fuzzy matching.

$x$. To achieve this, we identify matches for the end of $x$, $w_{:i-1}$, using a sliding window of size $k$ (Fig. 2(b)). We then count the occurrence of each token $w_i$ in the vocabulary set $\mathcal{V}$ following these matches and normalize to obtain the next-token probability:

$$P_{\text{induction}}^{(\text{fuzzy})}(w_{:i-1}w_i|x) = \frac{c_{\text{fuzzy}}(w_{i-k-1:i-1}w_i|x)}{\sum_{w_j \in \mathcal{V}} c_{\text{fuzzy}}(w_{i-k-1:i-1}w_j|x)}, \tag{3}$$

$$\text{where } c_{\text{fuzzy}}(w_{i-k-1:i-1}w_i|x) = \sum_{w_{j-k-1:j} \subset x} \mathbb{1}_{w_j = w_i} s_{\text{FM}}\left(w_{j-k-1:j-1}, w_{i-k-1:i-1}\right). \tag{4}$$

This similarity score serves as a floating count for the next token. In cases where the sequences $x_1$ and $x_2$ are exactly matched, as in the case of Infini-gram, we have $s_{\text{FM}}(x_1, x_2) = 1$, which is equivalent to increasing the count by one. The window size $k$ specifies the number of tokens to be considered in fuzzy matching.

### 3.3 Prediction of Generalized Induction head Model

By employing both the Infini-gram algorithm and Fuzzy Matching Model, GIM searches for the most relevant match—either exact or fuzzy—within the preceding tokens given a query at the end of the context (Fig. 3). Once a match is identified, it retrieves the token that followed the prior occurrence as the next-token prediction. By explicitly modeling this process, our method provides a transparent and controllable alternative to implicit in-context learning mechanisms in LLMs.

## 4 Results: Next-token Prediction for Language Modeling

### 4.1 Experimental Setup

**Datasets & evaluation** We use 4 text datasets for evaluation: BabyLM [68], OpenWebText [17], Pile [69], and FineWeb ([70]; `sample-10BT` subset), using some as the reference corpus and some as test datasets (Table 1). When testing, we report performance on 100k sequences randomly sampled with a context length of 1024 and a stride of 512 [14, 27].[4] We evaluate next-token prediction via accuracy, i.e. whether the top-predicted token was the correct token.[5]

**Baselines** We compare against Infini-gram as our sole baseline, as it is the state-of-the-art n-gram model and the only fully interpretable model with token-level attribution for generation. We found that it consistently outperformed prior interpretable language models, e.g., a standard 5-gram model based on OpenWebText achieves 26.4% accuracy in next-token prediction on the Pile-val, lower than Infini-gram's 27.1%. For a detailed discussion of interpretable frameworks, please refer to Sec. 2.

---

[4]The BabyLM test set contains fewer than 100k sequences, yielding approximately 32k and 34k cases for the GPT-2 and LLaMA-2 tokenizers, respectively.

[5]We do not use perplexity, as the sparse next-token predictions from n-gram models often assign zero probability to the top-ranked token, resulting in undefined or extremely high perplexity scores [14].

Table 1: Next-token prediction accuracy (%) for language modeling. Gray shading represents alignment between the reference corpus and the test dataset.

| Reference Corpus | | Model | Test Dataset | | |
|---|---|---|---|---|---|
| Type | # of Tokens | | BabyLM-test | FineWeb | Pile-val |
| **Tokenizer: GPT-2** | | | | | |
| - | - | GIM (exact) | 36.7 | 17.2 | 37.0 |
| - | - | GIM (fuzzy) | 41.1 | 25.2 | 38.7 |
| BabyLM-dev | 17.4M | Infini-gram | 37.6 | 14.7 | 16.0 |
| | | **+GIM** | 42.2 (+4.6) | 25.3 (+10.6) | 40.0 (+24.0) |
| Pile-val | 383M | Infini-gram | 16.6 | 20.1 | - |
| | | **+GIM** | 41.5 (+24.9) | 25.5 (+5.4) | - |
| OpenWebText | 9.04B | Infini-gram | 16.7 | 25.5 | 22.7 |
| | | **+GIM** | 41.8 (+25.1) | 27.2 (+1.7) | 42.7 (+20.0) |
| Unknown | ~10B | LLM (GPT-2) | 46.9 | 39.0 | 52.3 |
| **Tokenizer: LLaMA-2** | | | | | |
| - | - | GIM (exact) | 37.0 | 19.6 | 32.6 |
| - | - | GIM (fuzzy) | 42.7 | 28.3 | 38.5 |
| BabyLM-dev | 18.9M | Infini-gram | 39.0 | 17.1 | 13.2 |
| | | **+GIM** | 43.1 (+4.1) | 28.6 (+11.5) | 39.6 (+26.4) |
| Pile-val | 394M | Infini-gram | 19.0 | 24.1 | - |
| | | **+GIM** | 42.9 (+23.9) | 28.4 (+4.3) | - |
| OpenWebText | 10.3B | Infini-gram | 20.1 | 29.5 | 27.1 |
| | | **+GIM** | 43.2 (+23.1) | 30.3 (+0.8) | 42.1 (+15.0) |
| Pile-train | 383B | Infini-gram | 33.5 | 39.3 | 49.2 |
| | | **+GIM** | 49.4 (+15.9) | 38.0 (-1.3) | 50.3 (+1.1) |
| Unknown | ~2T | LLM (LLaMA2-7B) | 62.2 | 57.1 | 64.4 |

**Integrating GIM with Infini-gram**  For language modeling, we integrate GIM with Infini-gram, enabling the use of both reference corpus statistics and in-context distributions:

$$
P(y|x) = \begin{cases} P_{\infty}^{(\text{exact})}(y|x) & n_{\infty} > n_x \text{ and } n_{\infty} > \tau, \\ P_{\text{induction}}^{(\text{exact})}(y|x) & n_x \geq n_{\infty} \text{ and } n_x > \tau, \\ P_{\text{induction}}^{(\text{fuzzy})}(y|x) & \text{Otherwise,} \end{cases} \tag{5}
$$

where $n_{\infty}$ and $n_x$ are the effective $n$ when matching from a reference corpus or the input context, respectively. When these values are low, fuzzy matching is employed to compensate for the limited effective $n$. When the effective $n$ values from both the input context and reference corpus are equal, the estimate from the input context is prioritized. The hyperparameter $\tau$ determines how frequently exact matching is used over fuzzy matching; we set $\tau$ to 8 and 9 for the GPT-2 and LLaMA-2 tokenizers, respectively, based on cross-validation results (see Appendix A.3 for details).

## 4.2  Improving Next-token Prediction Accuracy with Contextualization

**Prediction performance of in-context matching**  GIM relies solely on the input context to predict the next token (limited to 1024 tokens in our evaluation). Table 1 shows that, despite this, GIM (exact) can outperform Infini-gram, which uses the OpenWebText dataset as a reference corpus, comprising approximately 10B tokens when tokenized with LLaMA-2 and 9.04B with GPT-2, by a margin of 5.5%p to 20%p on the BabyLM and Pile datasets. Infini-gram using BabyLM-dev as the reference corpus slightly outperforms GIM (exact) on the BabyLM-test, with performance gaps of 0.9%p and 2.0%p for the GPT-2 and LLaMA-2 tokenizers, respectively, under the aligned setting of reference corpus and input context. As shown in Fig. 4(a), Infini-gram (green) performs better in cases with a high effective $n$, even surpassing LLM (blue). However, significantly more cases have a low effective $n$ (histogram), where GIM (exact) (orange) outperforms Infini-gram. This finding underscores that in-context matching reflects the input query's distribution, leading to more accurate next-token predictions than reference matching, even when the reference corpus contains abundant tokens, especially under distribution shifts between the reference corpus and the test input.

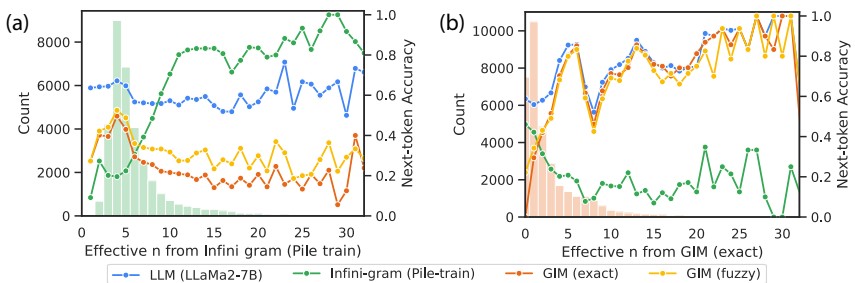

Figure 4: Comparison of next token prediction accuracy on BabyLM-test dataset, depending on effective $n$ from (b) Infini-gram and (b) GIM (exact). LLaMA-2 tokenizer is used.

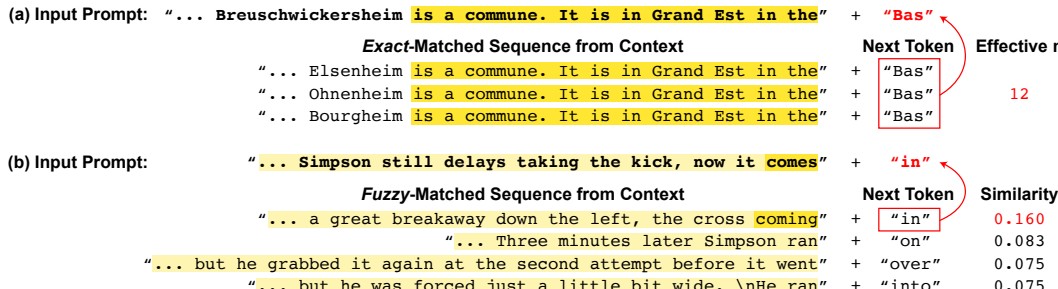

Figure 5: GIM's token-level attribution by tracing predictions to (a) exact or (b) fuzzy matches in prior context. Yellow highlight shows the match, and red box marks the source of the final prediction.

**Prediction improvements from GIM**   GIM (fuzzy), using Fuzzy Matching Model, consistently outperforms GIM (exact) with a margin of 1.7%p to 8.7%p (Table 1). This improvement is particularly evident in cases with low effective $n$. As illustrated in Fig. 4(b), the majority of cases within the input context have low effective $n$ (histogram), indicating that finding exactly matched long sequences within the limited amount of tokens is challenging. Fuzzy matching helps to provide better estimations for next-token predictions in these scenarios. Specifically, when the effective $n$ is less than 3, GIM (fuzzy) (yellow) demonstrates better performance than GIM (exact) (orange). Since many cases fall into this range, the overall accuracy of GIM (fuzzy) is higher.

The improvements achieved through the use of induction and fuzzy matching enable Infini-gram with GIM to outperform Infini-gram built on 383B tokens, improving performance by up to 15.9%p. While enlarging the reference corpus boosts performance, GIM offers a more efficient alternative to scaling from 10.3B to 383B tokens—a 38-fold increase. Moreover, GIM is a complementary approach that can be applied orthogonally to Infini-gram, regardless of the size of the reference corpus.

### 4.3   Qualitative Example of GIM Prediction

Fig. 5 shows examples of explanations provided by GIM. In the first case, the prompt exactly matches a 12-gram in the context, so GIM follows it to predict the next token. In the second, no exact match exists for the prompt ending in "comes", but GIM finds the most similar sequence ending in "coming" and follows it for prediction. These cases illustrate how GIM predicts from retrieved sequences, with transparency into which tokens contribute and how they are combined.

## 5   Results: Next-token prediction for fMRI Responses to Natural Language

Understanding how and where semantic information is represented across the human brain is a central objective in neuroscience. In this work, we extend prior modeling frameworks that learn mappings between natural language stimuli and corresponding neural responses across voxels, which are small three-dimensional regions of the brain. [71, 43]

## 5.1 Experimental Setup

We analyzed publicly available data[6] from [72] and [73], in which three human participants listened to 20+ hours of English-laguage podcast narratives while their fMRI responses were recorded across 95,556 cortical voxels. Our goal was to predict the brain response of each voxel from the language input heard by the participant[7]. We extracted text embeddings from the input story, then fit linear models to map these embeddings to fMRI responses on the training split (24 stories), and evaluated performance on the test split (2 stories) using bootstrapped ridge regression. Embeddings are extracted in various ways (described below) for each word in the input, and then interpolated to make predictions for the fMRI data that is

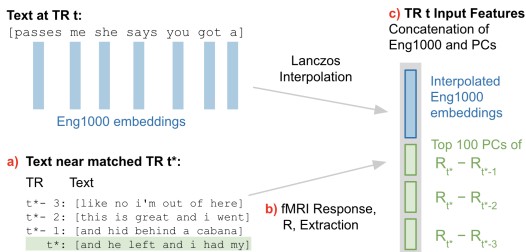

Figure 6: fMRI feature construction. (a) At each TR $t$, we retrieve a prior TR $t^*$ with various matching methods. (b) We extract the top 100 principal components (PCs) of neural response changes before $t^*$. (c) These are concatenated to interpolated Eng1000 embeddings for fMRI signal prediction at TR $t$.

recorded at 2-second time of repetition (TR) intervals. To model temporal delays in the fMRI signal, we add 4 time-lagged duplicates of the input features. See more fMRI details in Appendix A.6.

**fMRI prediction baselines**   We use Eng1000 as our primary baseline, the state-of-the-art interpretable model for predicting fMRI responses to narrative stories from a seminal study of language selectivity [71]. Each element in an Eng1000 embedding corresponds to a co-occurrence statistic with a different word. We also compare against LLaMA2-70B [74] embeddings, which achieve the highest performance on this task [75] but are not interpretable. LLaMA embeddings are extracted with a 16-word sliding window, using the final-layer embedding of the last token in each window.

**GIM for fMRI prediction**   We construct our GIM for fMRI by searching the preceding story text in an fMRI session for semantic matches and retrieving the changes in the recorded brain response that follows each match. Specifically, to predict the fMRI response, $R_t$, for the TR $t$, we first find the TR $t^*$ for which the text input yields the highest cosine similarity to the next-token distribution of the text input at TR $t-1$. Next, we isolate the change in fMRI responses following TR $t^*$: we take the difference in the top 100 principal components of the response $R_{t^*} - R_{t^*-1}$ and use them as features. To deal with potential time delays in the fMRI signal, we additionally concatenate these features with the top 100 principal components of $R_{t^*} - R_{t^*-2}$ and $R_{t^*} - R_{t^*-3}$. These features, along with the interpolated Eng1000 embeddings, form the full input to the linear model predicting fMRI response at TR $t$ (see Fig. 6). When constructing the GIM for fMRI, we search over the most recent 1024 words and their corresponding fMRI responses. To measure similarity between two texts, we use the predicted next-word distributions yielded by GIM (exact) in the input context ($P_{\text{induction}}^{\text{(exact)}}$ in Eq. (5)), which we call *GIM matching*.

**Baseline matching methods**   We compare GIM matching against three baseline matching strategies. First, we use the predicted next-word distributions yielded by exact n-gram matching in the 10B-token OpenWebText reference corpus ($P_{\infty}^{\text{(exact)}}$ in Eq. (5)), which we call *Infini-gram matching*. Second, *Random matching* selects a random preceding TR as a match. Third, *Naive n-gram matching* searches for an exact match to the most recent 4-word n-gram in the input context, without relying on predicted next-word distributions that our GIM matching method relies on. Table A6 shows additional experiments with fuzzy matching methods that show little performance gain, likely due to noise and temporal smoothing in fMRI signals that diminishes the advantage of fuzzy matching.

## 5.2 Prediction Improvements from GIM Matching

Table 2 shows the average correlation values across all voxels for each similarity model. Eng1000, the primary interpretable baseline, achieved a mean test correlation of 0.072. In contrast, GIM matching

---

[6] https://github.com/OpenNeuroDatasets/ds003020
[7] We report results for subject UTS03 due to high fMRI data quality, including superior repeatability, minimal motion, and strong encoding model performance [72].

Table 2: fMRI test prediction performance for different models. Black-box encodings use LLaMA-2. Error bars show 95% CI.

| Feature Model | Mean Correlation | |
| --- | --- | --- |
| | All Voxels | Top 10% Voxels |
| Eng1000 | $0.072 \pm 0.0004$ | $0.220 \pm 0.0012$ |
| + Random matching | $0.069 \pm 0.0003$ | $0.197 \pm 0.0012$ |
| + Naive ngram matching | $0.068 \pm 0.0003$ | $0.194 \pm 0.0012$ |
| + Infini-gram matching | $0.069 \pm 0.0003$ | $0.200 \pm 0.0012$ |
| **+ GIM matching** | $\mathbf{0.087} \pm \mathbf{0.0005}$ | $\mathbf{0.265} \pm \mathbf{0.0011}$ |
| Black-box encodings | $0.096 \pm 0.0005$ | $0.268 \pm 0.0013$ |

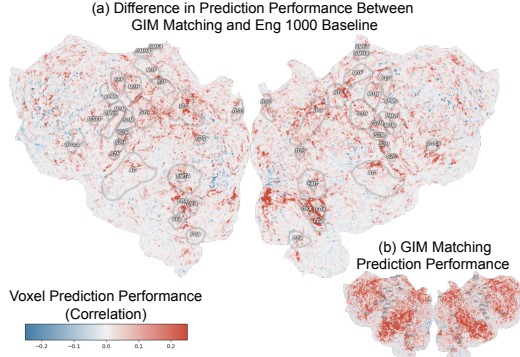

(a) Difference in Prediction Performance Between GIM Matching and Eng 1000 Baseline

(b) GIM Matching Prediction Performance

Voxel Prediction Performance (Correlation)

Figure 7: (a) Difference in the correlation performance between the GIM matching and the Eng1000 baseline, visualized across the cortex. (b) Correlation performance of GIM matching.

achieves a mean correlation of 0.087, a 20% improvement over Eng1000. When predicting the top-10% of voxels, GIM Matching achieves a mean correlation of 0.265, again a 20% improvement over Eng1000, and only 1% lower than the black-box LLaMA-2 model (mean correlation 0.268). In contrast, other matching-based baselines are unable to improve over Eng1000: The Naive n-gram matching baseline achieves a correlation of 0.068, and random matching achieves a correlation of 0.069.

Fig. 7 visualizes voxel-wise differences in test correlation performance between GIM matching and the Eng1000 baseline across the cortex. In line with prior studies linking model performance to functional localization [47, 45, 71, 43], GIM significantly improves prediction in regions highted in red such as the Occipital Face Area (OFA) and Intraparietal Sulcus (IPS). These gains reflect GIM's use of contextual input, in contrast to the static embeddings used by Eng1000, and suggest that these highlighted regions may contribute to contextual language processing.

**Describing improvements from GIM** To understand the improvements provided by matching, we summarize the text for inputs where each matching procedure (GIM and Infini-gram) performs well. We use an LLM to do the summarization, following recent works in LLM interpretability [76, 77]. We first identify phrases in the input story where a model's performance (average absolute error across voxels) exceeds the baseline performance by more than one standard deviation (see an example in Fig. 8). Then, we prompt GPT-4 ([2]; gpt-4-0613) to generate descriptions for these phrases.

Fig. 8 presents the unedited LLM descriptions[8]. GIM matching is described as capturing *Emotionally or Narratively Critical Phrases*, aligning with the idea that induction improves performance by tracking local context in a story, e.g., phrases that "are critical to the plot and character development". In contrast, Infini-gram matching is described as capturing *Brief, Stand-Alone Phrases*, matching the intuition that Infini-gram excels in capturing context that is not story specific, but "can stand alone with minimal context". To test these descriptions, we prompt GPT-4 to classify the identified phrases in two test stories using only the descriptions. This yields 61% accuracy, a moderate but significant improvement over chance (binomial test $p = 0.032$). See all phrases and prompts in Appendix A.6.

## 6 Discussion

GIM constitutes a significant step toward building mechanistically interpretable language models inspired by pre-trained LLMs. Unlike black-box models or partially interpretable approaches, GIM provides full transparency in next-token prediction while substantially narrowing the performance gap between interpretable and black-box architectures across two diverse domains. Importantly, GIM is not a general-purpose LLM or a tool to decode its internals; it isolates and reimplements a single observed capability, induction via repetition, as a fully interpretable module. This shows that high-performance behaviors implicitly learned by LLMs can be transparently reconstructed to advance

---

[8]Irrelevant preceding text such as "Sure here is the answer" is removed from the response.

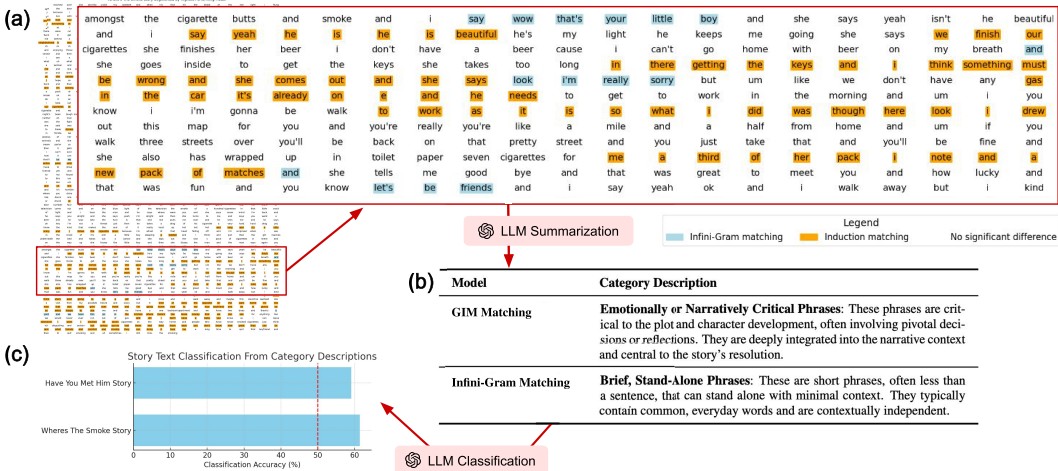

Figure 8: Qualitative illustration of how GIM and Infini-gram matching improve performance. (a) Highlighted phrases in the input story where the model outperforms the baseline. (b) Summary of the highlighted phrases by an LLM to characterize each matching method. (c) Classification of highlighted phrases in test stories based on the LLM-generated summaries.

performance in the interpretable modeling space. The transparency of GIM makes it well-suited for language modeling scenarios that require complete auditing, such as analyzing scientific texts or medical notes [78]. GIM's transparency also supports neuroscience research, as the fMRI analyses conducted here are a suggestive starting point for understanding how context is stored and recalled in the human cortex. GIM can further serve as a testbed for analyzing how context modulates the recall of specific semantic categories, like people and places, across the cortex, extending prior work with static embeddings [71]. Additionally, improvements from GIM Matching may help build encoding models that can more rapidly adapt to local context, which can be used in downstream applications such as brain decoding [73] or brain-computer interfaces [79].

GIM shows limited gains when the input context is short or uninformative. Its modular design enables the available context to be expanded through retrieval-augmented generation [80] or external memory. Like kNN-LMs [81], GIM's n-gram-based reasoning also struggles with tasks requiring deeper reasoning. Future work may explore hybrid approaches that pair GIM with black-box models for better trade-offs. Our speculative decoding setup, where GIM serves as a transparent draft generator verified by a larger LLM (Appendix A.5), illustrates one example in this direction. Another promising direction is expanding GIM beyond induction heads, integrating additional mechanistic components such as indirect object identifiers [42], numerical representations [82], retrieval heads [80], iteration heads [83], concept-level induction heads [84], instruction-following heads [85], or interpretable LLM submodules [86–88]. Finally, GIM's ability to model context-dependent patterns makes it well-suited for other sequential domains that require interpretability, e.g., it could be extended to study long-range dependencies in electronic health records [89], audio/speech models [90, 91], genomics [92] or financial time-series analysis [93].

## Acknowledgements

We would like to thank Lucas Liu, Ziyang Wu, and Paul Smolensky for insightful discussions and feedback throughout this work, which significantly contributed to the development of our ideas. This work was supported by Institute of Information & communications Technology Planning & Evaluation (IITP) grant funded by the Korea government (MSIT) [No.RS-2021-II211343, Artificial Intelligence Graduate School Program (Seoul National University), RS-2022-II220959, No.RS-2025-02263754, Human-Centric Embodied AI Agents with Autonomous Decision-Making], the BK21 FOUR program of the Education and Research Program for Future ICT Pioneers, Seoul National University in 2025, the National Research Foundation of Korea (NRF) grant funded by the Korea government (MSIT) (No. 2022R1A3B1077720, 2022R1A5A708390811), and InnoCORE program of the Ministry of Science and ICT (1.250021.01).

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

# A Appendix

## A.1 Impact Statement

We introduce the Generalized Induction-head Model (GIM), which improves the performance of fully interpretable models while maintaining transparency, making interpretable models more viable for high-stakes applications. For example, in medical note generation, interpretable models can enhance transparency, enabling clinicians to audit AI-generated text and reduce the risk of hallucinations or biased outputs. Additionally, GIM's token-level grounding can improve fairness in language models and mitigate bias in automated decision-making. GIM also achieved significant speedups in speculative decoding compared to inference with LLaMA2-70B alone, making it suitable for deployment in compute-limited settings (see Appendix A.5). Despite these advantages, GIM does not fully close the performance gap with black-box models, particularly for tasks requiring extensive reasoning or broad world knowledge. Its reliance on input context may also limit effectiveness in some scenarios where high-quality data is not available.

## A.2 Training of Fuzzy Matching Model

**Architecture of Fuzzy Matching Model**  We train two Fuzzy Matching Models, one using the GPT-2 tokenizer and the other using the LLaMA-2 tokenizer. With GPT-2 tokenizer, Fuzzy Matching Model consists of four transformer layers, whereas it comprises three transformer layers when using LLaMA-2 tokenizer. Since relative position is crucial for calculating similarity, we incorporate Relative Positional Encoding [94], with a maximum relative position of 32 for the GPT-2 tokenizer and 64 for the LLaMA-2 tokenizer. The vocabulary embeddings are initialized with those from GPT-2 and LLaMA2-7B, ensuring that the number of heads and embedding dimensions align with the specifications of GPT-2 and LLaMA2-7B.

**Creating similarity pair with LLMs**  For both Fuzzy Matching Model, we use LLaMA2-7B as a teacher model. OpenWebText and Pile-train[9] datasets for training each Fuzzy Matching Model that use GPT-2 or LLaMA-2 tokenizer. During training, we randomly sample sequences of 32 or 64 tokens with batch size of 128 or 256, resulting in 4,096 or 16,384 next-token prediction probabilities per batch. From these, we sample distant 3,584 or 4,096 queries and 512 keys and create similarity pairs ($3,584 \times 512$ or $4,096 \times 512$) by calculating similarity based on Equation (5). The models are trained using a combination of CE loss and reverse KLD loss, with equal weights (1.0). We adopt most of the training settings from the codebase[10] for training. Gradients are accumulated over 16 iterations, and we use the AdamW optimizer [95] with a learning rate of 0.0001 and a weight decay of 0.1. The learning rate follows a cosine schedule with a warmup over the first 1,000 iterations, and training continues for 15,000 or 20,000 iterations. Training is conducted on four NVIDIA A100 GPUs.

**Ablation study on Fuzzy Matching Model training**  We conduct an ablation study on the positional encoding strategy and training process of Fuzzy Matching Model using the OpenWebText dataset to distill it from LLaMA-2-7B. The study evaluates the contributions of Relative Positional Encoding, reverse KLD loss, and CE loss to the model's effectiveness. As shown in Table A1, next-token prediction accuracy improves significantly when both reverse KLD and CE losses are included, demonstrating their complementary roles in optimizing the Fuzzy Matching Model. With CE loss, Forward KLD loss is less effective than reverse KLD loss. Furthermore, using Relative Positional Encoding instead of Sinusoidal Positional Encoding leads to better performance, highlighting the advantages of incorporating relative positional information for enhanced fuzzy matching capabilities.

We also perform an ablation study on the training data. Table A2 shows that the performance difference between OpenWebText and Pile-train is minimal. When trained on Pile-of-Law [96], a more domain-specific corpus, Fuzzy Matching Model exhibits slightly lower performance. This suggests that domain specificity may slightly limit the generalization ability of the fuzzy matching module. Nevertheless, the approach remains robust even with more domain-specific training data.

---

[9] https://huggingface.co/datasets/monology/pile-uncopyrighted
[10] https://github.com/karpathy/minGPT

Table A1: Ablation study on training of Fuzzy Matching Model. Next-token accuracy (%) of GIM (fuzzy) on the BabyLM-test is reported. LLaMA-2 tokenizer is used.

| Positional Encoding | Reverse KLD loss | Forward KLD loss | CE loss | Accuracy |
|---|---|---|---|---|
| Relative | ✓ | | ✓ | 43.2 |
| Relative | | ✓ | ✓ | 42.8 |
| Relative | | | ✓ | 42.7 |
| Relative | ✓ | | | 41.9 |
| Sinusoidal | ✓ | | ✓ | 37.0 |

Table A2: Ablation study on training Fuzzy Matching Model with different datasets. Next-token accuracy (%) of GIM (fuzzy) on the BabyLM-test is reported. LLaMA-2 tokenizer is used.

| Dataset | Accuracy |
|---|---|
| OpenWebText | 43.2 |
| Pile-train | 42.7 |
| Pile-of-law | 41.8 |

## A.3 Determination of $\tau$

To build GIM by integrating the three types of estimations, we first need to determine the threshold for effective $n$, denoted as $\tau$. To identify the optimal value of $\tau$, we conducted cross-validation using the BabyLM training set (100M tokens). BabyLM consists of six datasets: `open_subtitles`, `bnc_spoken`, `gutenberg`, `childes`, `simple_wiki`, and `switchboard`. Since `switchboard` contains only 2M tokens, we exclude it from the experiment. For the remaining datasets, we use each dataset as a validation set, while the other four are used as the reference corpus to build Infini-gram. We then compare the performance changes of Infini-gram, GIM (exact), and GIM (fuzzy) depending on effective $n$. 10k samples are used for evaluating on each dataset.

As shown in Figure A1, Infini-gram outperforms GIM (exact) when the effective $n$ exceeds 8 for the GPT-2 tokenizer and 9 for the LLaMA-2 tokenizer. Therefore, we set $\tau$ to 8 and 9 for the respective tokenizers.

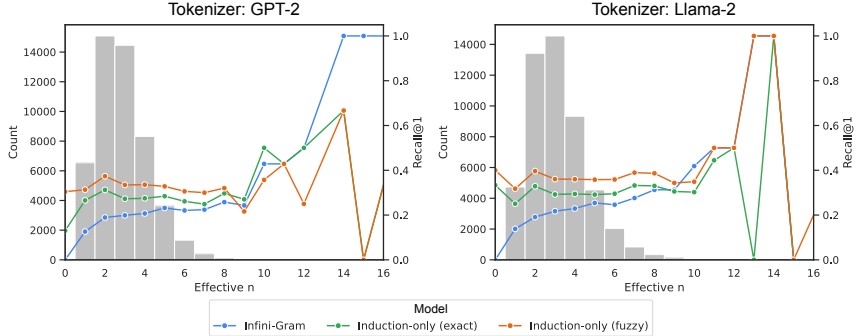

Figure A1: Comparison of next-token accuracy.

## A.4 Language Modeling Results Extended

**Experimental details**   We use diverse datasets as reference corpus for Infini-gram. We use Infini-gram that is released by authors[11] for Pile-train[12] and Pile-val[13]. For BabyLM-dev and OpenWebText,

---

[11] https://infini-gram.io/api_doc.html
[12] `v4_piletrain_llama`
[13] `v4_pileval_llama` and `v4_piletrain_gpt2`

Table A3: Ablation study on components of **Infini-gram with GIM**. Next-token accuracy (%) on BabyLM-test is reported.

| Reference Corpus | BabyLM-dev | Pile-val | OpenWebText | Pile-train |
|---|---|---|---|---|
| Infini-gram with GIM | 43.1 | 42.9 | 43.2 | 49.4 |
| w/o GIM (fuzzy) | 42.2 | 36.9 | 38.3 | 46.6 |
| w/o GIM (exact) | 43.0 | 42.8 | 43.1 | 49.3 |
| w/o Infini-gram | | | 42.9 | |
| Infini-gram | 39.0 | 19.0 | 20.1 | 33.5 |

Table A4: Speed of speculative decoding (SP). Accept. denotes the acceptance rate (%). The mean and standard deviation of 3 runs are reported.

| | Draft Model | Large Model | SP | BabyLM-test Accept. rate (%) | Speed ms/token (↓) | Up (↑) | Pile-val Accept. rate (%) | Speed ms/token (↓) | Up (↑) |
|---|---|---|---|---|---|---|---|---|---|
| A40×1 | | LLaMA2-7B | | | 30.2±0.0 | | | 30.2±0.1 | |
| | TinyLLaMA-1.1B | LLaMA2-7B | ✓ | 78.7±0.5 | 21.3±0.0 | 1.42 | 78.3±0.1 | 21.3±0.6 | 1.42 |
| | GIM (fuzzy) | LLaMA2-7B | ✓ | 74.9±1.1 | 17.7±0.7 | 1.71 | 71.2±0.5 | 20.1±0.4 | 1.50 |
| | | LLaMA2-13B | | | 52.4±0.0 | | | 52.0±0.2 | |
| | TinyLLaMA-1.1B | LLaMA2-13B | ✓ | 78.2±0.0 | 26.7±0.5 | 1.96 | 77.6±0.1 | 26.3±0.5 | 1.98 |
| | GIM (fuzzy) | LLaMA2-13B | ✓ | 73.5±0.1 | 24.8±0.1 | 2.11 | 69.8±0.2 | 27.8±0.1 | 1.87 |
| H100×2 | | LLaMA2-13B | | | 26.4±0.1 | | | 26.3±0.4 | |
| | LLaMA2-7B | LLaMA2-13B | ✓ | 78.9±0.0 | 24.7±0.0 | 1.07 | 78.6±0.0 | 25.1±0.3 | 1.05 |
| | TinyLLaMA-1.1B | LLaMA2-13B | ✓ | 78.3±0.1 | 20.7±0.1 | 1.28 | 77.6±0.1 | 21.5±0.1 | 1.22 |
| | GIM (fuzzy) | LLaMA2-13B | ✓ | 73.2±0.3 | 13.3±0.2 | 1.98 | 69.9±0.1 | 14.9±0.1 | 1.77 |
| | | LLaMA2-70B | | | 71.2±0.1 | | | 71.0±0.2 | |
| | LLaMA2-7B | LLaMA2-70B | ✓ | 77.2±0.2 | 38.3±0.5 | 1.86 | 77.8±0.2 | 37.4±0.3 | 1.90 |
| | TinyLLaMA-1.1B | LLaMA2-70B | ✓ | 75.5±0.1 | 35.3±0.2 | 2.02 | 76.3±0.4 | 33.9±0.6 | 2.10 |
| | GIM (fuzzy) | LLaMA2-70B | ✓ | 68.5±0.6 | 31.4±0.7 | 2.27 | 66.6±0.6 | 33.3±0.6 | 2.13 |

we build our own Infini-gram. We use public code to build and inference Infini-gram[14] and GIM (exact)[15]. During inference, the maximum length for exact matching with Infini-gram is 500, and we use window size $k$ for fuzzy matching as 32 and 64 for GPT-2 and LLaMA-2 tokenizers, respectively.

**Ablation study on Infini-gram with GIM** We conduct an ablation study to assess the impact of each component in Infini-gram with GIM. Table A3 reports next-token accuracy when individual components are omitted. Excluding GIM (fuzzy) results in a more significant performance drop than removing GIM (exact). This underscores the importance of fuzzy matching in handling diverse contexts and improving adaptability, as reflected in Table 1, where GIM (fuzzy) outperforms GIM (exact). Since both components act as induction heads, they exhibit complementary roles—when one is removed, the other partially compensates for its absence. Only when using Pile-train as a reference corpus, omitting Infini-gram leads to the most substantial performance decline. It is worth noting that when the reference corpus lacks similarity to the test dataset's distribution (*e.g.*, Pile-val, OpenWebText, and Pile-train), the performance of Infini-gram falls significantly below the scenario where it is not utilized at all. This highlights the sensitivity of Infini-gram to the quality and relevance of the reference corpus.

## A.5 Speculative Decoding

GIM offers both interpretability and efficiency. When combined with LLMs in speculative decoding, it enhances prediction accuracy while significantly boosting inference speed.

---

[14] https://infini-gram.io/pkg_doc.html
[15] https://github.com/AlexWan0/infini-gram/tree/main

Table A5: Speed of speculative decoding (SP). The mean and standard deviation of 3 runs are reported.

| Draft Model | Large Model | SP | BabyLM-test | | Pile-val | | FineWeb | |
|---|---|---|---|---|---|---|---|---|
| | | | ms/token ($\downarrow$) | Speed Up ($\uparrow$) | ms/token ($\downarrow$) | Speed Up ($\uparrow$) | ms/token ($\downarrow$) | Speed Up ($\uparrow$) |
| | LLaMA2-13B | | 26.4±0.1 | | 26.3±0.4 | | | |
| GIM (fuzzy) | LLaMA2-13B | ✓ | 13.3±0.2 | 1.98 | 14.9±0.1 | 1.77 | 14.9±0.3 | 1.76 |
| GIM | LLaMA2-13B | ✓ | 23.1±0.4 | 1.14 | 22.8±0.3 | 1.15 | 23.0±0.7 | 1.14 |
| | LLaMA2-70B | | 71.2±0.1 | | 71.0±0.2 | | 71.1±0.2 | |
| GIM (fuzzy) | LLaMA2-70B | ✓ | 31.4±0.7 | 2.27 | 33.3±0.6 | 2.13 | 33.2±1.0 | 2.15 |
| GIM | LLaMA2-70B | ✓ | 42.0±0.7 | 1.70 | 41.6±1.0 | 1.71 | 40.4±1.2 | 1.76 |

**Experimental details**   To evaluate the efficiency of GIM (fuzzy), we compare the inference time for speculative decoding with TinyLLaMA[16] and LLaMA2-7B [74]. We evaluate speculative decoding by generating up to 1024 tokens, using a prefix of 1024 tokens. The speed of decoding may vary depending on the computational environment. To ensure robust evaluation across different setups, we conduct experiments in two environments: one with a single NVIDIA A40 GPU and 128 CPU cores, and another with two NVIDIA H100 GPUs and 64 CPU cores. Greedy sampling is used for token generation, and each experiment is repeated three times with different random seeds.

**Induction improves speculative decoding performance**   Table A4 demonstrates the speed-up effect of speculative decoding with GIM (fuzzy). GIM (fuzzy) relies solely on the induction power derived from the input context to predict the next token, leading to lower acceptance rates compared to LLMs. Despite this, its inference speed is remarkably fast, and it often matches the predictions of large models. As a result, the speed improvement can exceed $2\times$ compared to using LLaMA2-70B alone. In most cases, GIM (fuzzy) achieves even greater speed gains than when using an LLM as a draft model for speculative decoding.

Additionally, we would like to note that speculative decoding with GIM (fuzzy) and a pretrained LLM not only accelerates the inference speed of the pretrained model but also enables explainable predictions based on the given input context. When accurate predictions can be made through interpretable methods, we utilize this process for interpretability. In more challenging cases, we rely on a larger model that, while less interpretable, delivers better performance for accurate predictions. Thus, this approach provides a balanced method that addresses both interpretability and accuracy, in addition to enhancing efficiency.

Table A5 reports the inference times for GIM (fuzzy) and GIM using speculative decoding, with the OpenWebText dataset serving as the reference corpus for Infini-gram. We find matches with a maximum of 64 tokens for both exact and fuzzy matching. The experiments are conducted on two NVIDIA H100 GPUs and 64 CPU cores. Although GIM requires more time for generation on average than GIM (fuzzy), it still significantly reduces inference time compared to relying solely on a large model for inference.

**Explanation**   Figure A2 presents several examples of explanations provided by GIM. Even if an exact match fails to yield a good match, when the probability of subsequent tokens is similar, the fuzzy matching model can predict with high similarity, enabling successful fuzzy matching, enabling successful fuzzy matching, and improving next-token prediction.

## A.6   fMRI results extended

**Data details**   We analyze publicly available data collected from prior work [72, 73]. Methods from the previous study are summarized here for completeness. Functional magnetic resonance imaging (fMRI) data was recorded from three healthy participants as they listened to English-language podcast stories over Sensimetrics S14 headphones. Participants were only instructed to listen to the stories. No explicit behavioral responses were required.

For the collection of training data, each participant completed approximately 20 hours of listening sessions across 20 separate sessions in which unique stories were presented. This produced 33,000

---
[16]https://huggingface.co/TinyLLaMA/TinyLLaMA-1.1B-intermediate-step-1431k-3T

**GIM (exact)**

**(a) Input Prompt:** "... `_Frontispiece_--(_Page 61_)] \nBUNNY BROWN AND HIS`      **"ER"**

| Sequence from Context | Effective n | Next Token |
|---|---|---|
| "... `PG70358 = = = \nBUNNY BROWN AND HIS SIST`" | 13 | ER |

**(b) Input Prompt:** "... `Then the chorus: "Will you, won't you, will you, won'`"      **"t"**

| Sequence from Context | Effective n | Next Token |
|---|---|---|
| "... `out in a friendly voice:\n"Will you, won't you, will you, won'`" | 13 | t |

**GIM (fuzzy)**

**(c) Input Prompt:** "... `Because he says it's Lincolnshire ! \nNo, he didn't! \nHe said`"      **"it"**

| Sequence from Context | Similarity | Next Token |
|---|---|---|
| "... `What's Lincolnshire gotta do with it? \nBecause he says`" | 0.680 | it |
| "... `God that wind's gone cold! \nI say`" | 0.210 | that |
| "... `Well he don't know anything about gardening, you see! \nBut`" | 0.203 | I |
| "... `What's Lincolnshire gotta do with it? \nBecause`" | 0.186 | he |
| "... `I don't know why`" | 0.179 | ! |

**(d) Input Prompt:** "... `So I taught him that the first week, and the second`"      **"week"**

| Sequence from Context | Similarity | Next Token |
|---|---|---|
| "... `And I was running it and the first`" | 0.098 | week |
| "... `who's erm sixty odd and he comes in here every`" | 0.087 | day |
| "... `And I was running it and the first week I got there, and one`" | 0.053 | gu |
| "... `So I taught him that the first`" | 0.042 | week |
| "... `we had to cancel because nobody turned up.\nEr one`" | 0.035 | of |

Figure A2: Examples of explanation of GIM from BabyLM-test. (a) and (b) show examples of exact matching while (c) and (d) show examples of fuzzy matching. The red box marks the source of the final prediction.

timepoints per voxel across the entire human cortex. For testing data collection, participants heard two-held-out stories five times each and a third story ten times (one story per session). These repeated measurements were averaged to improve reliability. Signal-to-noise ratios for each voxel were estimated using the mean-explainable variance approach from [97]. Analysis was restricted to voxels that were located within 8mm of the cortical mid-surface, yielding about 90,000 voxels per participant.

All participants were healthy adults with normal hearing and gave written informed consent. The study protocol was approved by the Institutional Review Board of the University of Texas at Austin. The scans were acquired on a 3 T Siemens Skyra MRI system at the University of Texas at Austin using a 64-channel Siemens head coil. Functional images were obtained with a gradient-echo EPI sequence (TR = 2.0 s, TE = 30.8 ms, flip angle = 71°, multi-band factor = 2, voxel size = 2.6 mm × 2.6 mm × 2.6 mm, matrix size = 84 × 84, field of view = 220 mm). Anatomical scans were collected using a T1-weighted multi-echo MP-RAGE sequence with 1 mm isotropic voxels, following the standard Freesurfer morphometry protocol [98].

Functional data was preprocessed with FSL 5.0 using the FMRIB Linear Image Registration Tool (FLIRT) for motion correction and alignment. Each participant's runs were registered to a subject-specific template built from the first functional run of the first session, with all automated registrations manually verified for accuracy. Low-frequency signal drifts were removed using a second-order Savitzky–Golay filter with a 120 s window. To mitigate onset artifacts and detrending issues near scan boundaries, 20 s (10 volumes) were discarded from both the start and end of each run. This eliminated the initial silent period and the first and last 10 s of each story. Each voxel's mean response was then subtracted, and the remaining signal was normalized to unit variance.

To ensure temporal alignment between linguistic and neural data, word onset times were interpolated to the fMRI sampling rate using Lanczos interpolation with a window size of 3. The hemodynamic response was modeled as a finite impulse response (FIR) with four time lags ($-8$, $-6$, $-4$, and $-2$ s), following the approach of [71]. For every subject $x$, voxel $v$, we fit an encoding model $g_{(x,v)}$ to

Table A6: fMRI Prediction Performance when using fuzzy matching. Error bars show 95% CI.

| Feature Model | Tokenizer | Matching Model | Mean Correlation | |
|---|---|---|---|---|
| | | | All Voxels | Top 10% Voxels |
| Eng1000 | - | - | $0.072 \pm 0.0004$ | $0.220 \pm 0.0012$ |
| Infini-gram + Eng1000 | GPT-2 | - | $0.069 \pm 0.0003$ | $0.200 \pm 0.0012$ |
| GIM Matching + Eng1000 | GPT-2 | - | $0.087 \pm 0.0005$ | $0.265 \pm 0.0011$ |
| Fuzzy Induction Matching + Eng1000 | GPT-2 | GPT-2 | $0.076 \pm 0.0004$ | $0.222 \pm 0.0011$ |
| Fuzzy Induction Matching + Eng1000 | LLaMA-2 | LLaMA2-70B | $0.076 \pm 0.0004$ | $0.225 \pm 0.0012$ |
| Fuzzy Induction Matching + Eng1000 | GPT-2 | Fuzzy Matching Model | $0.076 \pm 0.0004$ | $0.216 \pm 0.0011$ |
| Fuzzy Induction Matching + Eng1000 | LLaMA-2 | Fuzzy Matching Model | $0.077 \pm 0.0004$ | $0.223 \pm 0.0012$ |

predict the BOLD response $\hat{B}$ from the embedded stimulus, i.e. $\hat{B}_{(x,v)} = g_{(x,v)}(H_i(\mathcal{S}))$. Model evaluation was performed on the held-out test stories, using the trained encoding models to predict and assess voxel responses.

**fMRI fuzzy induction head settings**    Similar to the GIM Matching technique described in Sec. 5.1, we construct an induction head for fuzzy matching. In the fuzzy setting, we leverage the predicted next-word distributions obtained through fuzzy n-gram matching in the input context ($P_{\text{induction}}^{(\text{fuzzy})}$ in Equation (3)), which we refer to as *Fuzzy Induction Matching*. Specifically, we calculate the cosine similarity between the next-word distributions of the current word and all prior candidate words.

To account for the temporal resolution of fMRI, we apply Lanczos smoothing to the word-level similarity values, aligning these values with the fMRI time scale. This allows us to identify the time point (TR) $t^*$ that maximally corresponds to the current time point $t$ based on the highest similarity.

We evaluate several configurations for deriving the next-word distributions, including GPT-2, LLaMa-2, the Fuzzy Matching model with the GPT-2 tokenizer, and the Fuzzy Matching Model with the LLaMA-2 tokenizer. See more details on Fuzzy Matching models in Sec. 3.2.

**Extended prediction performance results**    The prediction performance of Fuzzy Induction Matching Models is compared to the performance of the GIM Matching Models and the Eng1000 baseline in Table A6. The Fuzzy Induction Model, in its highest-performing configuration (using the Fuzzy Matching Model with the LLaMa2-70B tokenizer), achieves only a 6.94% improvement in prediction performance compared to the Eng1000 baseline. The lower relative performance of Fuzzy Induction Matching compared to GIM Matching may be due to the inherent noise and lower spatial and temporal resolution of fMRI data, which makes it challenging to detect subtle differences in neural activations associated with similar but non-identical stimuli.

Table A7: GPT-4 Prompts for Generating and Classifying Categories of Text. Ellipses (...) indicate omitted portions of the full prompts.

| Title | Prompt |
|---|---|
| GPT-4 Prompt for Generating Category Descriptions | I have provided two test stories below. Specific phrases from each story have been picked out based on the performance of different encoding models. Can you describe the characteristics of the words and phrases that each category contains? Be specific about the type of words, their context in the story, and any other relevant commonalities. Write succinct descriptions for each category that would allow one to categorize phrases in other such stories accurately.
Category A: ['sh first she digs into her cutoffs in the', 'both need this right now i', ... ]
Category B: ['to everything or you make yourself scarce', 'my cigarettes and uh', ...]
Full Story: [['i reached over and secretly'], ['undid my seatbelt'], ...] |
| GPT-4 Prompt for Classifying Stages Based on Descriptions | I have attached category descriptions below. Based on the descriptions, in order, go through each short list of words (short phrase) in the story at the end and classify the segments into one of the categories. Rather than listing all the phrases in a category at a time, list each phrase in order and label it as belonging to category A or B.
Category A: Emotionally, or Narratively Critical ...
Category B: Brief, Stand-Alone Phrases ...
Full Story: [['i reached over and secretly'], ['undid my seatbelt'], ...] |

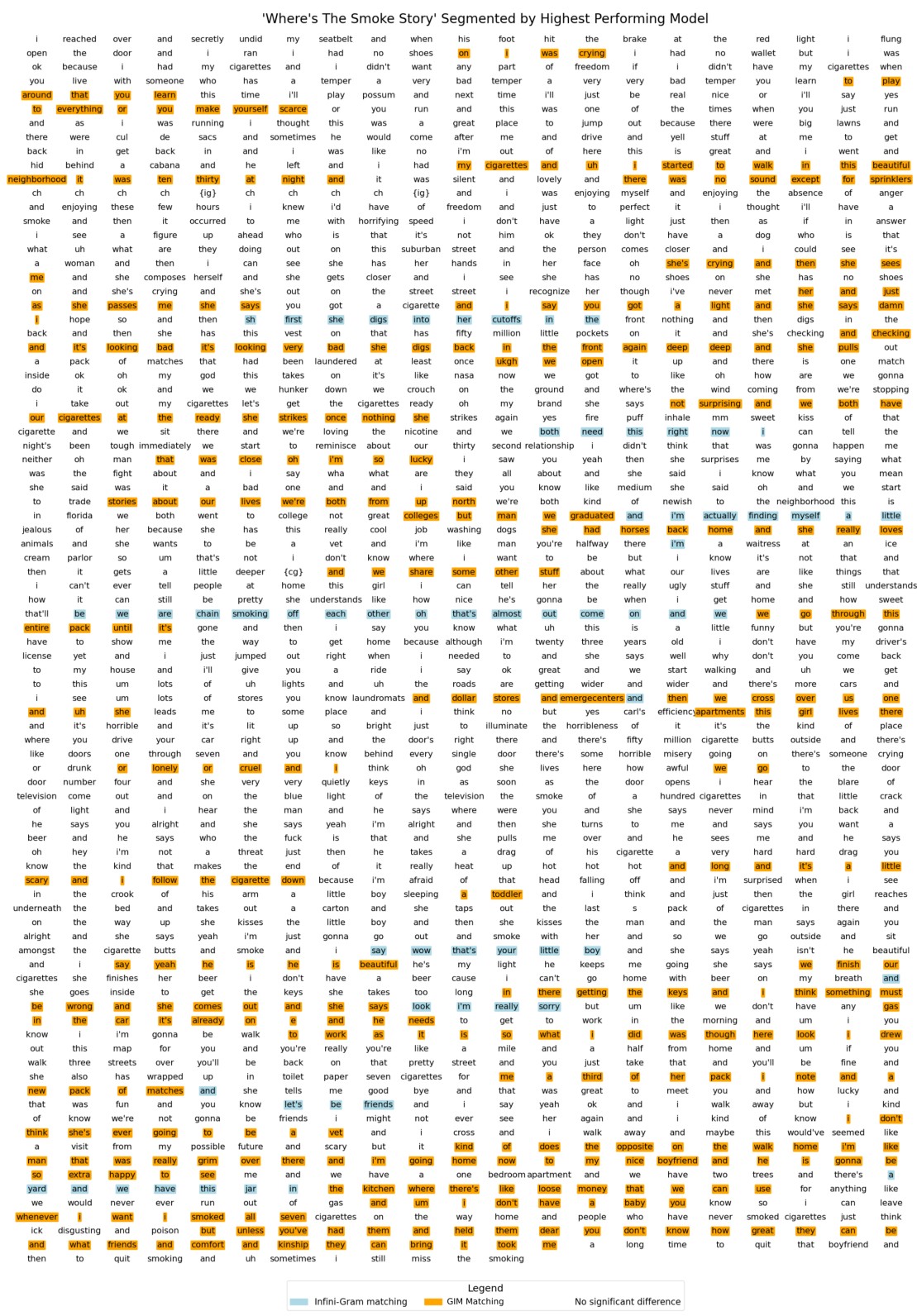

Figure A3: Test story 1 (*Where's There's Smoke*), highlighted in regions where the Infini-Gram matching and GIM matching models exceed baseline performance, measured by the average absolute error across voxels, by more than one standard deviation.

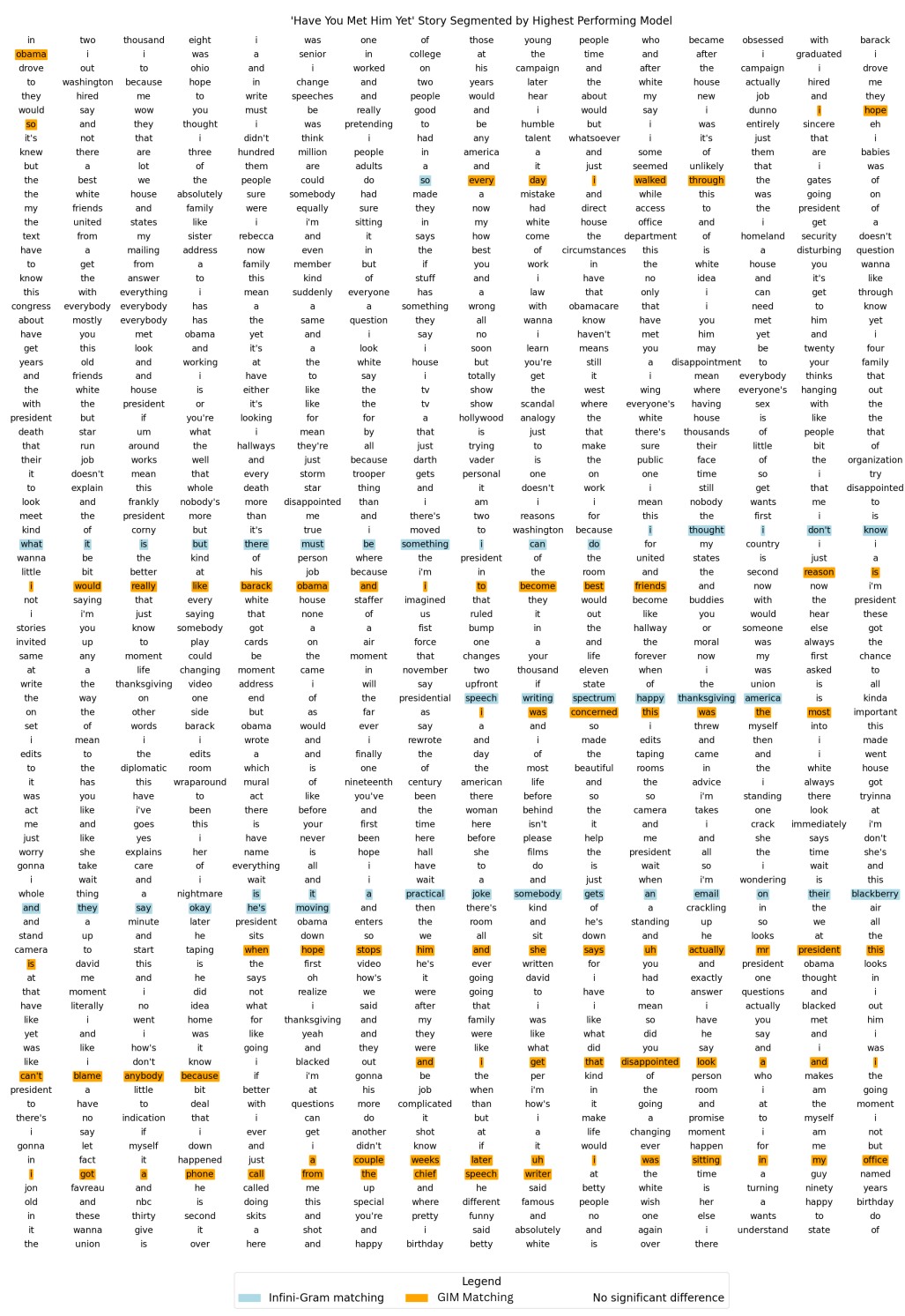

Figure A4: The first section of test story 2 (*Have You Met Him Yet*), highlighted in regions where the Infini-Gram and GIM matching models exceed baseline performance.

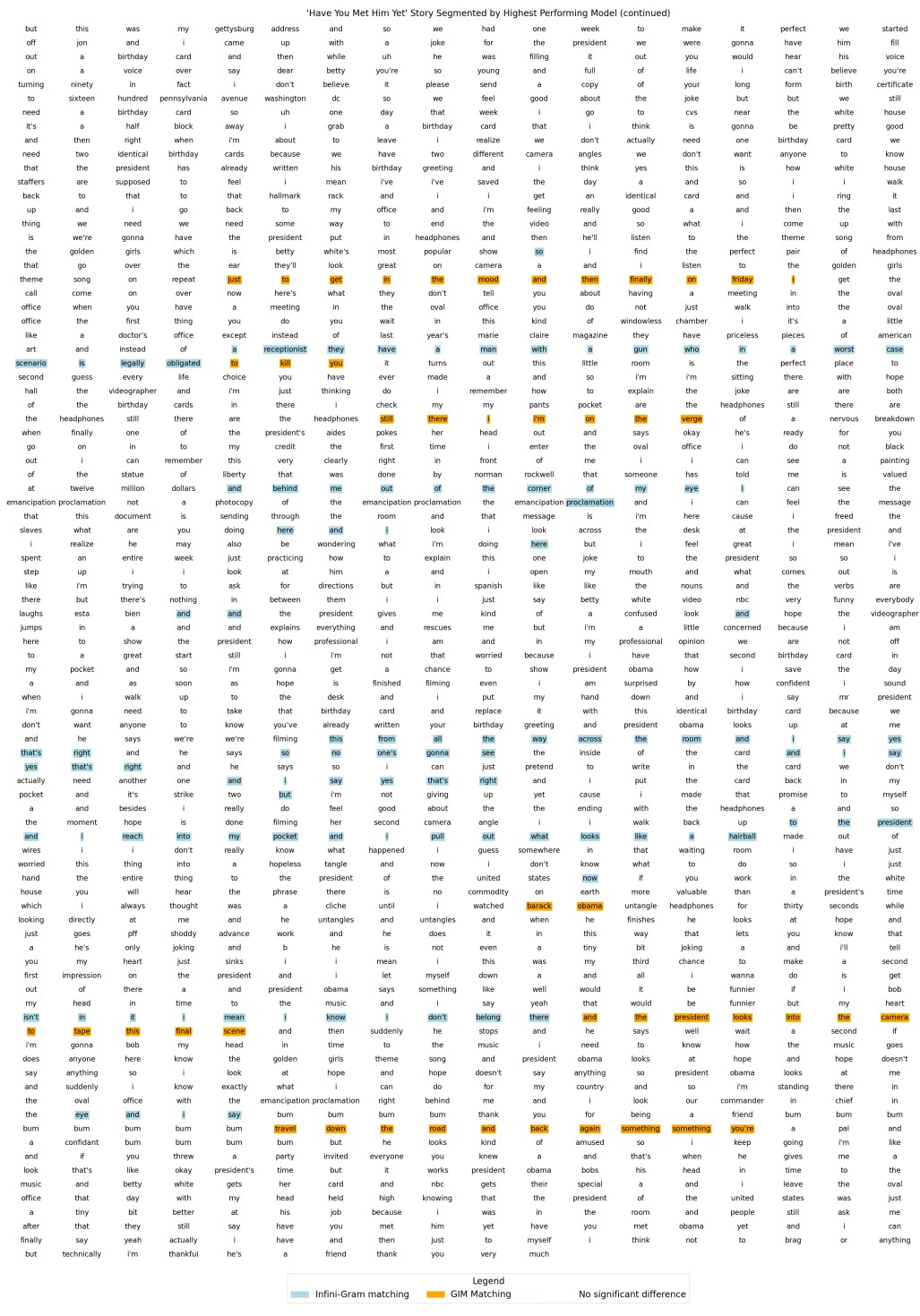

Figure A5: The second section of test story 2 (*Have You Met Him Yet*), highlighted in regions where the Infini-Gram and GIM matching models exceed baseline performance.

