# OpenReview forum: "Interpretable Next-token Prediction via the Generalized Induction Head"
_NeurIPS.cc/2025/Conference — NeurIPS 2025 poster_

### Official Review · Reviewer_WRxr · 2025-06-27

**Clarity:** 3
**Significance:** 3
**Originality:** 3
**Rating:** 5
**Confidence:** 3

**Summary:**

This paper introduces the Generalized Induction-Head Model (GIM), an interpretable framework for next-token prediction.  By integrating exact n-gram matching with fuzzy neural similarity matching, GIM overcomes the limitations of traditional models.  It achieves up to 25% higher next-token prediction accuracy than interpretable baselines in language modeling and a 20% improvement in fMRI response prediction.  GIM bridges the gap between interpretability and performance, providing transparent predictions  into language processing mechanisms.

**Questions:**

1. Is there any structural relationship between the GIM and the induction heads in traditional transformers? Can GIM provide interpretability for prediction mechanisms in real-world large language models?
2. Which component of the matching strategy - the n-gram matching or the fuzzy neural similarity matching - plays a decisive role in the accurate prediction of language models? Specifically, how does the hybrid design of these two strategies help improve the prediction accuracy and robustness of the model for input changes?
3. Given GIM's context length limitation, which improvement direction - such as dynamic window scaling, hierarchical context aggregation, or external memory integration - is most promising to enhance the model's generalization ability for long-range dependencies and sparse input contexts?

**Ethical Concerns:**

["NO or VERY MINOR ethics concerns only"]

**Final Justification:**

The clarification that the context-length limitation was a choice for evaluation rather than an inherent architectural constraint of GIM is a crucial point that resolves my primary reservation about long-range dependencies.

Furthermore, the reframing of GIM not as a tool for post-hoc analysis of LLMs, but as a standalone, interpretable implementation of a specific mechanism (induction) makes the paper's contribution much clearer. The proposed additions to the discussion section will be valuable for readers. The new empirical results demonstrating the fuzzy matching component's robustness on out-of-domain data are also very convincing.

**Limitations:**

yes

**Quality:**

4

**Strengths And Weaknesses:**

Strengths
1. The paper introduces the Generalized Induction-Head Model (GIM), a novel framework that integrates exact n-gram matching with fuzzy neural similarity metrics to enable transparent next-token prediction.
2. This paper highlights GIM's interpretability through exact n-gram matching, which overcomes the black-box nature of traditional transformers.

Weaknesses
1. GIM’s exact n-gram matching relies on fixed-size sliding windows (e.g., 1024 tokens), which restricts its ability to capture long-range dependencies. When input contexts are insufficient to contain informative matches, the model’s effective n drops, leading to performance degradation, highlighting limitations in modeling extended contextual understanding.
2. GIM is designed as a task-specific framework for next-token prediction rather than a generalizable model to decode LLM internal mechanisms. This framework does not directly address how pre-trained transformers (e.g., GPT-2) generate predictions. For example, GIM’s hybrid matching does not clarify whether LLM success stems from explicit pattern retrieval or implicit semantic compression.
3. The fuzzy matching module, trained via knowledge distillation from LLMs, may introduce semantic ambiguity when input distributions deviate from training data. While the neural similarity metric improves robustness to minor variations, it might struggle with out-of-domain inputs.

---

> ### Author Rebuttal · Authors · 2025-07-30
>
> We thank the reviewer for their thoughtful and encouraging feedback. We appreciate the recognition of (1) its transparency through n-gram-based reasoning and (2) its potential for interpretable next-token prediction. Below, we respond to the reviewer’s concerns.
>
> ### W1. Fixed-size window and long-range dependency
> We agree that the use of fixed-size sliding windows limits long-range retrieval, but this is a design choice for evaluation, not an inherent constraint of the GIM framework. In fact, GIM (and its predecessor Infini-gram) is designed to construct n-gram and fuzzy-matching indices over the entire available context. For example, when using OpenWebText, we treat each full document as a single context and build an Infini-gram over the entire span. This enables GIM to effectively consider contexts that extend to millions or billions of tokens, limited only by hardware. Thus, the performance drop under short contexts reflects a general property of all context-based models, not a structural limitation of GIM itself.
>
> ### W2. GIM’s scope and relation to LLM internals
> GIM is not intended to decode the full internal mechanisms of pretrained LLMs. Instead, as described in Lines 109–118, it reimplements a specific, well-studied behavior—induction via repetition—as a standalone, fully interpretable module. The goal is not to analyze LLMs post hoc, but to show that a core capability observed in them can be recreated faithfully and transparently.
>
> While GIM may offer a useful reference for understanding induction-like behavior in LLMs (pattern copying from prior input spans), it does not aim to explain their full predictive capabilities, which rely on many additional mechanisms such as compression, memorization, and reasoning. Rather than serving as a diagnostic tool, GIM demonstrates how targeted mechanisms can be isolated, audited, and extended through interpretable design. As discussed in the Related Work section, GIM builds on both classical n-gram modeling and recent findings in mechanistic interpretability, but is distinct in offering a fully transparent generation process for next-token prediction.
>
> We will clarify this distinction more explicitly in the revised Discussion section, emphasizing that GIM is a composable interpretability unit, not a general-purpose LLM substitute, as follows:
>
> Updates to the first paragraph of the Discussion (Lines 286-289):
> > GIM constitutes a significant step toward building mechanistically interpretable language models inspired by pre-trained LLMs. Unlike black-box models or partially interpretable approaches, GIM provides full transparency in next-token prediction while substantially narrowing the performance gap between interpretable and black-box architectures across two diverse domains. Importantly, GIM is not a general-purpose LLM or a tool to decode its internals; it isolates and reimplements a single observed capability, induction via repetition, as a fully interpretable module. This shows that high-performance behaviors implicitly learned by LLMs can be transparently reconstructed to advance performance in the interpretable modeling space.
>
> Updates to the second paragraph of the Discussion (Lines 298-302):
> > GIM shows limited gains when the input context is short or uninformative. Its modular design enables the available context to be expanded through retrieval-augmented generation [76] or external memory. Like kNN-LMs [77], GIM’s n-gram-based reasoning also struggles with tasks requiring deeper reasoning. Future work may explore hybrid approaches that pair GIM with black-box models for better trade-offs; our speculative decoding setup, where GIM serves as a transparent draft generator verified by a larger LLM (Appendix A.5), illustrates one example in this direction.
>
> ### W3. Robustness of fuzzy matching under distribution shift
>
> To empirically evaluate semantic ambiguity in fuzzy matching under domain shift, we trained the Fuzzy Matching Model on Pile-of-Law[a], a domain-specific legal corpus that differs significantly in both style and content from the general-domain corpora (e.g., OpenWebText). When evaluated on the Pile-val benchmark, the performance drops by only 1.4%p.
>
> |  | OpenWebText | Pile of Law |
> |---|---|---|
> | GIM (fuzzy) | 43.2 | 41.8 |
> | Infini-gram + GIM | 49.8 | 48.7 |
>
> This 1.4%p gap indicates that the learned similarity metric remains robust even under substantial distributional shift. Moreover, because GIM combines fuzzy matching with an exact n-gram component—which is distribution-agnostic—the overall prediction performance remains largely stable. These results suggest that the hybrid design (Infini-gram + GIM) preserves accuracy and interpretability even for out-of-domain inputs for neural similarity metric.
>
> [a] Henderson. et al. "Pile of law: Learning responsible data filtering from the law and a 256gb open-source legal dataset." NeurIPS (2022).
>
> ### Q1. Structural relation between GIM and LLM induction heads
> GIM directly implements the behavior of induction heads observed in LLMs in an explicit and auditable way. While induction heads in LLMs emerge implicitly through training, GIM constructs this mechanism by design, allowing for faithful, fully traceable attribution. GIM is not intended to serve as a tool for reverse-engineering LLM internals; instead, it distills one known mechanism into a standalone, interpretable module (as we mentioned in our response to W2 above).
>
> GIM may serve as a useful reference point for understanding induction-like behavior in black-box models. For example, one could compare the predictions of GIM and those of a large LLM on a controlled task to evaluate whether the LLM exhibits similar input-output dependencies, potentially offering a lightweight probe into the presence or strength of induction-head-like activity (for a detailed comparison, please refer to our response to Reviewer 3WB2, **W2: Comparison to attribution-based interpretability**). Still, our goal with GIM is not to interpret full LLM behavior, but to demonstrate that interpretable modules can replicate specific high-performance behaviors seen in large models.
>
>
> ### Q2. Role of exact vs. fuzzy matching
> Both n-gram (exact) and fuzzy (neural similarity) matching play important and complementary roles in GIM, but fuzzy matching tends to be more decisive for overall performance and robustness.
>
> As shown in Table 1 (Lines 190–195), GIM (fuzzy) consistently achieves higher accuracy than GIM (exact) across datasets. Similarly, Table A3 (Appendix Lines 599–605) shows that removing the fuzzy component from the full model (i.e., Infini-gram + GIM (exact), the row for ‘w/o GIM (fuzzy)’) results in a greater performance drop than removing the exact component, indicating the stronger standalone utility of fuzzy matching.
>
> However, we view the hybrid design as key to GIM’s overall performance and adaptability. Exact n-gram matching provides high-precision retrieval when long repeated spans are available, enabling accurate token prediction based on exact recurrence. Fuzzy matching enhances robustness to minor variations such as typos, rephrasings, or paraphrases, especially when exact matches are sparse. As illustrated in Figure 4(b), fuzzy matching performs better when the effective n is small, i.e., when long exact matches are unavailable. In contrast, when high-effective-n matches exist, exact matching tends to be more effective. GIM dynamically selects between these estimates based on match quality and effective n (Equation (5)), enabling it to maintain high performance even under input perturbations or distribution shifts.
>
> This hybrid approach ultimately allows GIM to balance accuracy and robustness, by accurately modeling repetition when it occurs while generalizing gracefully in its absence (e.g., under distribution shifts or in low-redundancy contexts).
>
> ### Q3. Future directions for scaling context length
> We appreciate the reviewer’s suggestions. Before addressing them directly, we would like to reiterate a key point discussed in our response to **W1 (Fixed-size window and long-range dependency)**: if the concern is about the length of the considered context, analogous to how LLMs use long prompts, then GIM does not face an inherent limitation. In our implementation, GIM is applied over the full available context (e.g., an entire OpenWebText document), which may span millions of tokens. Its context length is constrained only by available hardware, not by the model’s design.
>
> If the concern instead pertains to modeling higher-level abstractions or long-range semantic dependencies beyond surface-level repetition, we agree this is an important direction for future work. Among the proposed options—dynamic window scaling, hierarchical context aggregation, and external memory integration—we view external memory as the most promising extension. GIM’s retrieval-based nature makes it a natural fit for augmentations like memory banks or document caches, which could expand its effective range while preserving transparency and token-level traceability. We already note in the Discussion (Line 299) that retrieval-augmented generation (RAG) is a natural future direction for GIM, as it would similarly expand context by integrating external knowledge sources.
>
> If we have misunderstood the reviewer’s intent, we would be glad to revisit this question with further clarification.

---

> > ### Comment · Reviewer_WRxr · 2025-08-06
> > **Official Comment by Reviewer WRxr**
> >
> > Thank you to the authors for the detailed and thorough rebuttal, which has effectively addressed my main concerns. I appreciate the clarification that GIM's context-length limitation is an evaluation choice rather than an inherent architectural constraint, as this resolves my primary reservation regarding long-range dependencies. Furthermore, repositioning GIM as a standalone, interpretable implementation of the induction mechanism, rather than a tool for post-hoc LLM analysis, makes the paper's contribution much clearer. Given that my major concerns have been resolved, I have raised my score accordingly.

---

> > > ### Author Response · Authors · 2025-08-06
> > >
> > > We sincerely thank the reviewer for their careful reading of our rebuttal and for the thoughtful follow-up. We’re pleased that our response helped address the reviewer’s concerns—particularly those regarding the context-length setting and the intended scope of GIM. We greatly appreciate the constructive feedback throughout the process.

---

### Official Review · Reviewer_3WB2 · 2025-06-29

**Clarity:** 3
**Significance:** 3
**Originality:** 3
**Rating:** 4
**Confidence:** 4

**Summary:**

The paper introduces the Generalized Induction-Head Model (GIM), a fully interpretable architecture for next-token prediction inspired by "induction heads" found in large language models (LLMs). Unlike black-box LLMs, GIM offers transparency by retrieving similar input sequences using both exact and fuzzy matching, enabling direct attribution of predictions to input context. It combines traditional n-gram approaches with a learned neural similarity metric to match sequences even under minor variations. GIM significantly improves prediction accuracy in both language modeling and neural response prediction tasks.

**Questions:**

- Provide more details of the Fuzzy Matching Model.
   1. Line 131 mentions that it consists of a “few transformer layers”. How many?
   2. Line 131 mentions, it is “designed to output feature embeddings.” Please provide more details on what exactly is the output feature embeddings and how are they computed.
 - Could you describe the $w_{:i-1}$ notation in some detail?
 - Line 180 mentions the 10B-token OpenWebText dataset. However, Table 1 reports it as 9.04B. Why the discrepancy?
 - Line 181 mentions that GIM (exact) outperforms infini-gram by a margin of 5.5% to 20%. I’m unsure where is 5.5% coming from?
 - In Table 1, why is tokenizer information under Type column? Additionally, could you describe what does reference corpus mean in the context of GPT-2 and LLaMA-2-7B?
 - Line 238 mentions next-token distribution. I’m unsure what exactly it means in the fMRI setting. I understand text is converted to embedding and then a linear model is trained for each voxel. But how is this next-token distribution computed?
 - It would also be helpful if you could describe the response variable in the fMRI setting. What exactly is this response variable and how is it used to compute the principal components?
 - Line 302-305 mentions incorporating other mechanistic interpretability insights to improve GIM. How do you envision doing it?

**Ethical Concerns:**

["NO or VERY MINOR ethics concerns only"]

**Final Justification:**

As mentioned in my rebuttal response, my primary concern with the current draft is its framing. Although the authors stated in the rebuttal that GIM represents a fundamentally different paradigm, focusing primarily on interpretability rather than performance, the current version of the manuscript suggests otherwise.

**Limitations:**

The paper already notes GIM’s limited improvement when the input context is short. Beyond that, its performance is likely to be poor on symbolic reasoning tasks such as variable binding. For example, given the following code snippet, GIM would likely predict 5 as the next token instead of the correct value 15:
`x = 5; x += 10; x =`

**Quality:**

2

**Strengths And Weaknesses:**

**Strengths**
 - It builds on insights from the mechanistic interpretability literature to develop a better language model, rather than limiting those insights to controlled or toy settings.
 - It integrates Infini-gram with GIM to effectively leverage contextual information and enhance next-token prediction accuracy.
 - It evaluates the effectiveness of the proposed GIM model not only for language modeling but also for fMRI response prediction.

**Weaknesses**
 - Although GIM achieves comparable next-token prediction accuracy to GPT-2 over long contexts, I'm not confident it would perform similarly to GPT-2 on broader benchmarks such as MMLU, code, or math. These tasks are crucial, as many real-world applications of language models rely heavily on strong performance in these areas.
 - While the paper compares GIM’s performance to baselines like Infini-gram, it does not compare it to pretrained language models analyzed using interpretability techniques to identify vital tokens. For instance, we can estimate the importance of an input token using the derivative of output w.r.t that token. It would be valuable to see whether the contextual information identified by GIM could also be uncovered through such methods.
 - Some parts of the manuscript could be improved; please refer to the Questions section for details.

While the idea of augmenting n-gram language models with contextual information is compelling, it's unclear why one would choose GIM over pretrained models. The main advantage seems to be GIM's ability to attribute next-token predictions to the existing corpus and/or context. However, similar attribution can be achieved using interpretability techniques applied to pretrained models. Therefore, it’s important to compare both approaches in terms of interpretability and evaluate their effectiveness more comprehensively. Such a comparison would help determine whether the proposed GIM is practically viable.

---

> ### Author Rebuttal · Authors · 2025-07-30
>
> We thank the reviewer for their thoughtful and constructive review. We are especially grateful for the recognition of (1) extending mechanistic interpretability insights, (2) integrating Infini-gram with GIM to leverage context, and (3) validating the model across language and neuroscience domains.
>
> ### W1. Role of GIM
> We agree that tasks such as MMLU, code generation, and math require strong reasoning skills, which benefit from the broad parametric capacity of large black-box LLMs. However, GIM represents a fundamentally different paradigm: it is designed to maximize interpretability, not to replicate the full capabilities of LLMs or serve as a stand-alone replacement.
>
> Our contribution is to show that a large fraction of next-token prediction—particularly repetition-based induction—can be captured by a fully transparent mechanism, substantially narrowing the performance gap between interpretable and black-box models (see Tables 1 and 2). This marks a meaningful step for interpretable language modeling.
>
> At the same time, GIM is explicitly composable to bridge transparency and general-purpose capability. As noted in Appendix A.5, we already demonstrate a hybrid approach where GIM acts as a transparent draft generator and a larger LLM serves as a verifier via speculative decoding. This design preserves interpretable predictions for accepted tokens while retaining the strengths of large models. We also refer the reviewer to our response to Reviewer WRxr, **W2: GIM’s scope and relation to LLM internals**, where we further clarify this distinction and outline our revision plan.
>
> ### W2. Comparison to attribution-based interpretability
>
> We appreciate the reviewer’s suggestion to compare GIM with attribution-based methods on pretrained models. While both GIM and attribution techniques may highlight context tokens that influence a prediction, they differ fundamentally in motivation, mechanism, and reliability.
>
> Post-hoc attribution methods, such as gradients or relevance propagation, estimate token importance after a parametric model makes a prediction. They rely on heuristic approximations of causal influence and are sensitive to implementation choices, often failing faithfulness tests [a]. This issue is particularly prominent in NLP, where interactions between many tokens make it difficult for attribution methods to reliably interpret the importance of individual tokens.
>
> In contrast, GIM is interpretable by design. It does not approximate influence but generates predictions through explicit retrieval of matched n-grams. Every unit of probability mass is directly traceable to specific spans in the input. This makes GIM not just a model with interpretable outputs, but a model with a transparent and causal prediction mechanism.
>
> We agree that comparing the two approaches is meaningful, as GIM is designed to model a key mechanism in LLMs. Such a comparison helps assess whether GIM’s transparent behavior reflects how LLMs handle contextual repetition.
>
> We compared GIM’s matched sequences with LXT [b] attributions from LLaMA-2-7B. For LXT, we display tokens with attribution scores ≥ 0.10 or in the top 10, annotated as _`WORD[ATTRIBUTION VALUE]`_. Non-italic tokens have omitted attribution values. For GIM (exact), we underline the matched n-gram; for GIM (fuzzy), we report the top-3 highest similarity sequences with their scores. Note that GIM provides explanations at the sequence level, not token-by-token. Below are two examples:
>
> ### Example 1)
> **Query:** `"... I said that they don’t hang"`  -> **Prediction:** `"about"`
> - **LXT:** _`"<s>[0.33] … \n[0.59]`_`… she don’t`_`hang[0.10] about[0.36]`_`… she hangs`_`about[0.10]`_`… we don’t`_`hang[0.17] about[0.36]`_`… Billy`_`’[0.10]`_`s got …`_`they[0.18] don[0.14]’[0.10]t[0.20] hang[1.00]"`_
> - **GIM (exact):**  The last sequence _`"don’t hang"`_ matched with `"she`_`don't hang`_`about"` and `"we`_`don't hang`_`about"`
> - **GIM (fuzzy):**
>   - _`"… saying that we don't hang [0.32]`_`about"`
>   - _`"… But she don't hang [0.30]`_`about"`
>   - _`"… And then, hang [0.21]`_`on"`
>
> ### Example 2)
> **Query:** `"... They use it as an"`  -> **Prediction:** `"exc"`
> - **LXT:** _`"<s>[0.33]`_`… they’ve got`_`.[0.37]`_`… They’ll use it as`_`an[0.08] exc[0.22] use[0.21]. … They[0.08]`_`use it`_`as[0.26] an[1.0]"`_
> - **GIM (exact):** The last sequence _`"use it as an"`_ matched with `"They'll`_`use it as an`_`exc"`
> - **GIM (fuzzy):**
>   - _`"… They'll use it as an [0.73]`_`exc"`
>   - _`"… Cos if you left that with the [0.01]`_`first"`
>   - _`"… point out to them of course that the [0.01]`_`second"`
>
> Across examples, both GIM and LXT tend to focus on repeated patterns from earlier in the context—consistent with known induction head behavior (Lines 109–118). In this sense, attribution maps partially recover the same inductive patterns in LLMs that GIM is explicitly designed to model. However, key differences remain: attribution maps may assign high importance to non-semantic tokens (e.g., `\n`, `.`), and they require interpretation to infer causality. In contrast, GIM uses its retrieved sequences to construct next-token distributions, executing a faithful, copy-like mechanism.
>
> We recognize the reviewer’s concern that similar explanations might be achievable through interpretability techniques on pretrained models, but these remain approximate and offer no guarantees of faithfulness. GIM, by contrast, constructs predictions directly from retrieved spans, making attribution causal, controllable, and repeatable. This is not just a theoretical distinction, it is essential in domains like clinical decision-making, scientific modeling, or regulatory auditing, where interpretability must be built into the model, not added afterward (see Introduction, Lines 15–19).
>
> We will clarify this point in the revised Introduction (Lines 28–30), emphasizing that GIM is not a tool to interpret opaque systems, but an inherently interpretable system. We see these two approaches as complementary, not interchangeable.
>
> [a] Zhao. et.al. “Explainability for Large Language Models: A Survey,” ACM TIST (2024).
>
> [b] Achtibat. et.al. “AttnLRP: Attention-Aware Layer-Wise Relevance Propagation for Transformers,” ICML (2024).
>
> [c] Rudin. “Stop explaining black box machine learning models for high stakes decisions and use interpretable models instead,” Nature Machine Intelligence (2019).
>
> ### W3. Clarity improvements
> We respond to each question below.
>
> **Q1.** As described in Appendix A.2, the Fuzzy Matching Model contains 3 Transformer encoder layers (LLaMA-2 tokenizer) or 4 (GPT-2 tokenizer). It outputs embedding vectors from the final layer without projection. These embeddings are used for cosine similarity (Eq. 2).
>
> **Q2.** The notation $w_{:i−1}$ refers to the full input up to but excluding token $w_i$, which denotes the final token. Thus, $w_{:i−1}$ forms the query context for GIM matching.
>
> **Q3.** The discrepancy in token counts stems from tokenizer differences. OpenWebText contains ~10B tokens with LLaMA-2, and ~9.04B with GPT-2. We will clarify this in Lines 180–181.
>
> **Q4.** This refers to the evaluation on Pile-val with the LLaMA-2 tokenizer: GIM (exact) 32.6 vs. Infini-gram 27.1 (Δ = 5.5%p).
>
> **Q5.** The “Type” column refers to model type (e.g., GIM (exact), GIM (fuzzy)), not to the tokenizer. The table is divided into two blocks: the upper block uses the GPT-2 tokenizer, and the lower block uses the LLaMA-2 tokenizer. The table is split into two blocks: GPT-2 (upper) and LLaMA-2 (lower). Since tokenizers segment text differently, the same corpus yields different token counts (e.g., OpenWebText = 9.04B with GPT-2 vs. 10.3B with LLaMA-2), which can impact performance.
>
> **Q6.** The “next-token distribution” refers to the probability distribution over next tokens computed by GIM based on the story text the subject has heard up to each time point (TR). In the fMRI setting, we use this distribution as a proxy for internal model state: for a given TR t, we find a prior TR t* whose GIM distribution is most similar (by cosine similarity), and retrieve the change in fMRI response that followed hearing the text at t*. This allows us to link similar linguistic contexts, according to GIM, to similar brain responses. We have clarified this in the main text (Lines 238–239).
>
> **Q7.** The response variable refers to the fMRI recordings themselves: specifically, the BOLD signal measured across 95,556 voxels at each 2-second time repetition (TR). To reduce dimensionality and extract dominant patterns of neural activity, we compute principal components over the voxel responses across time. The changes in these PCs following matched linguistic contexts are then used to construct GIM-based features, capturing how the brain’s response evolves in response to similar text input. We will clarify this in the revised paper.
>
> **Q8.** GIM explicitly models induction heads—key mechanisms in LLMs for in-context learning—as interpretable modules. We believe more such modules can be designed for other known behaviors. We envision building a more general and transparent language model by composing such modules. For example, future work could incorporate circuits for indirect object identification as standalone modules, or introduce retrieval heads that explicitly fetch external knowledge when needed. These modules could be implemented using interpretable retrieval schemes or rule-based operations similar to GIM, with each module contributing a distinct, traceable component to the next-token distribution. As interpretability research progresses, GIM-style components may serve as the building blocks of transparent yet capable models.
>
> ### L1. Symbolic reasoning limitations
> We agree. Tasks requiring symbolic reasoning or variable binding are beyond GIM’s current scope. It lacks internal state tracking or arithmetic capacity. We discuss this limitation in the Discussion (Lines 300–302), and view this as a potential direction for future work.

---

> > ### Comment · Reviewer_3WB2 · 2025-08-03
> >
> > > However, GIM represents a fundamentally different paradigm: it is designed to maximize interpretability, not to replicate the full capabilities of LLMs or serve as a stand-alone replacement.
> >
> > I didn’t get this impression while reading the paper. The focus seemed oriented towards both interpretability and performance, specifically, building a model that is both interpretable and performant. For instance, Lines 41–43 state: *We integrate GIM into Infini-gram, and GIM improves next-token prediction accuracy by 25%p over Infini-gram using OpenWebText [17] as a reference corpus, significantly **narrowing the performance gap with GPT-2**.* Additionally, Lines 50–51 mention: *These results challenge the assumption that interpretability and predictive performance are fundamentally at odds.*
> >
> >
> > Thank you for sharing the new experimental results and for elaborating on the key differences between GIM and attribution-based methods. I agree that while they may appear similar on the surface, important distinctions exist. For example, attribution-based methods often assign high scores to non-semantic tokens, which complicates their interpretability. In the examples you provided, LXT tends to attribute importance to the beginning-of-string `<s>` token, a token that lacks inherent interpretability. This does not seem to be the case with GIM. Moreover, as you noted, there are empirical differences between post-hoc interpretability and models that operate in a functionally interpretable manner, especially relevant for high-stakes applications.
> >
> >
> > I also appreciate your vision for developing inherently transparent language models that incorporate empirical insights from the interpretability literature. This direction could enhance GIM’s ability to support variable binding, an essential component of symbolic reasoning, arithmetic, and factual recall.
> >
> >
> > While I still have some reservations about the overall framing of the work, I don’t believe that alone warrants rejection. Accordingly, I am raising my score.

---

> > > ### Author Response · Authors · 2025-08-05
> > >
> > > We sincerely thank the reviewer for carefully reading our rebuttal and providing thoughtful follow-up comments. We’re glad to hear that our response helped address the reviewer’s initial concerns, especially those regarding the performance gap with black-box LLMs and the comparison with post-hoc attribution methods.
> > >
> > > To avoid any potential misunderstanding, we will revise the main text to ensure that references to performance improvements do not obscure the paper’s main focus. Our goal was not to suggest parity with black-box LLMs, but to demonstrate that meaningful gains in predictive performance can be achieved without sacrificing interpretability—specifically, by outperforming existing interpretable models like Infini-gram. While our model does not yet match black-box LLMs in absolute performance, we wish to emphasize that GIM represents a step forward by narrowing this gap within a fully transparent framework.
> > >
> > > We again appreciate the reviewer’s thoughtful and constructive feedback, and we look forward to incorporating these clarifications into the final version of the paper.

---

### Official Review · Reviewer_WuPU · 2025-07-02

**Clarity:** 3
**Significance:** 3
**Originality:** 3
**Rating:** 5
**Confidence:** 2

**Summary:**

This paper proposes the Generalized Induction-Head Model (GIM), an interpretable approach to next-token prediction designed to address the lack of transparency in large transformer models. GIM integrates exact n-gram matching with a fuzzy retrieval mechanism based on embedding similarity. The model is evaluated on language modeling and fMRI neural response prediction tasks. It shows improved performance over interpretable baselines and narrows the performance gap with black-box models. Unlike its closest baseline, Infini-gram, the proposed model performs matching solely within the input context, without relying on an external corpus. As the authors demonstrate, this approach can improve performance, since in-context matching better aligns with the distribution of the input query.

**Questions:**

1. How sensitive are the results to the use of Jensen–Shannon divergence in the fuzzy matching procedure? Were other divergence measures, such as KL divergence, considered or evaluated?
2. While Jensen–Shannon divergence is introduced formally in equation (1), the implementation appears to rely on cosine similarity between embeddings. Could the authors clarify how the formulation in (1) maps onto the practical implementation in (2)?
3. In equation (4), what is the purpose of the indicator function? It seems the summation is not necessary if we only look for cases where w_j = w_i.

**Ethical Concerns:**

["NO or VERY MINOR ethics concerns only"]

**Final Justification:**

This paper presents an innovative technique and has a very thorough evaluation. I believe it's a solid contribution to the field.

**Limitations:**

Yes.

**Paper Formatting Concerns:**

No concerns.

**Quality:**

3

**Strengths And Weaknesses:**

While I am not familiar with the n-gram literature for LLMs, I found GIM to be a thoughtful and well-motivated extension of ideas from Infini-gram. Conceptually, it's quite similar—both aim to predict the next token based on prior context—but GIM takes a distinct approach by restricting retrieval strictly to the input context, which makes it more scalable and predictions more aligned to the input query distribution as demonstrated in Figure 4.

Another contribution of this paper is the fuzzy matching mechanism, in contrast to Infini-gram’s reliance on exact n-gram matches which requires a large external corpus for better matching. This makes GIM more robust to small changes in phrasing or noise, such as typos, and gives it more flexibility in practice. Surprisingly, even without access to an external corpus, GIM outperforms Infini-gram, which speaks to the effectiveness of using in-context information alone.

I also appreciated the paper's extension of the model to fMRI prediction tasks. This demonstrates GIM’s potential beyond standard language modeling, particularly in exploring how meaning is represented in the brain.

That said, I do see limitations. Because GIM depends solely on the input context for retrieval, its performance might not scale well in open-ended or highly diverse tasks where context is sparse or uninformative. This trade-off between interpretability and generalization is important and the authors could explore it more directly in future work.

---

> ### Author Rebuttal · Authors · 2025-07-30
>
> We thank the reviewer for their thoughtful and detailed feedback. We particularly appreciate the recognition of our paper’s contributions in three areas: (1) extending Infini-gram via in-context-only retrieval, (2) introducing fuzzy matching for improved robustness, and (3) validating the model across language and neuroscience domains.
>
> ### W1. Limitations of in-context-only retrieval
> We thank the reviewer for highlighting this consideration. While we agree that sparse or uninformative contexts can limit any model’s predictive ability, we do not view GIM’s reliance on input context as fundamentally at odds with generalization. In fact, many generalization behaviors in LLMs rely on identifying patterns in recent input [a, b]. GIM captures this same ability in an explicit and transparent way.
>
> GIM models this inductive behavior through both exact and fuzzy matching, enabling it to generalize across local variations even without access to large external corpora. As our results show, it often outperforms Infini-gram trained on massive datasets, demonstrating stronger generalization, especially in domains with repetitive or structured language. When richer or longer contexts are available, GIM benefits accordingly. It mirrors how black-box LLMs perform better when provided with more examples.
>
> Importantly, “input context” in GIM is not restricted to the literal text string passed at inference time. Its modular retrieval-based architecture allows it to seamlessly incorporate any additional information source, including external memory stores, retrieval-augmented generation (RAG) modules, or hybrid pipelines. For example, as we show in Appendix A.5, GIM can serve as a fast, interpretable draft generator in speculative decoding, with a larger model verifying predictions, demonstrating that accuracy and interpretability can be jointly achieved through composition.
>
> We’ll clarify this perspective more directly in the revised discussion (please see our response to Reviewer WRxr, **W2: GIM’s scope and relation to LLM internals**), and hope this clarifies GIM’s design intent and extensibility across contexts.
>
> [a] Olsson, Catherine, et al. "In-context learning and induction heads." arXiv preprint arXiv:2209.11895 (2022).
>
> [b] Brown, Tom, et al. "Language models are few-shot learners." NeurIPS (2022).
>
> **Q1.** We chose JSD over other measures (such as KL divergence) primarily due to its symmetry. Unlike KLD, JSD is symmetric, which aligns with our use case: the fuzzy-matching model must learn the similarity between two n-gram next-token distributions that stand on equal footing, rather than comparing an “estimated” distribution to a “true” one. Because the two distributions are treated symmetrically, a symmetric metric like JSD is the appropriate choice.
>
> **Q2.** Equation (1) defines the teacher-side similarity using JSD over next-token distributions. However, our fuzzy matching model—a student model trained via knowledge distillation—does not need to generate full token distributions during inference. Instead, it produces embedding vectors for n-grams, and the similarity between them is computed via cosine similarity (Equation (2)). During training, the student is supervised to match the similarity scores given by the teacher (based on JSD), effectively learning to embed sequences such that their cosine similarity reflects the similarity of their predictive distributions. This formulation allows us to bypass full next-token prediction at inference time while still preserving the semantics of the JSD-based similarity.
>
> **Q3.** In Equation (4), we sum over all candidate spans $w_{j−k−1:j}$ in the input context and $w_j$ can be different from $w_i$. The indicator function $\mathbb{1}_{w_j = w_i}$ ensures that we only include cases where the token following the matched n-gram ($w_j$) is equal to the specific token $w_i$ being scored. This allows us to accumulate a “floating count” of support for each token $w_i$ based on the similarity between preceding contexts.
>
> We appreciate the reviewer’s attention to both the conceptual framing and technical details, and we hope our responses adequately address your concerns.

---

> > ### Comment · Reviewer_WuPU · 2025-08-06
> > **Thank you**
> >
> > Thank you for addressing my questions with so much detail. I still believe this paper is a solid contribution to the field thus I am keeping my score.

---

### Official Review · Reviewer_vuy2 · 2025-07-03

**Clarity:** 3
**Significance:** 3
**Originality:** 2
**Rating:** 4
**Confidence:** 4

**Summary:**

This paper tackles the trade-off between performance and interpretability in language models by introducing the Generalized Induction-Head Model (GIM). Drawing inspiration from “induction heads” observed in large language models, GIM is a retrieval-based, interpretable model that operates entirely within the input context. It combines two core components: an exact n-gram matcher that flexibly identifies the longest possible match (rather than relying on a fixed n), and a fuzzy matcher based on a neural network to handle variations. GIM complements models like Infini-gram by offering richer in-context retrieval, helping address the limitations of external corpus-based search.

Empirically, GIM closes much of the gap with black-box LLMs: it improves next-token prediction accuracy by up to 25% over interpretable baselines and boosts fMRI response prediction by 20%. The model also proves useful for speculative decoding, enabling lossless acceleration of LLMs. Overall, this work shows how insights from reverse-engineering LLM internals can be repurposed to build transparent models that remain competitive—challenging the notion that interpretability must come at the expense of performance.

**Questions:**

Have you considered using GIM as a layer or integrating it as an attention head, similar to how the N-gram head was implemented in [1] ?

[1] https://arxiv.org/abs/2401.12973

**Ethical Concerns:**

["NO or VERY MINOR ethics concerns only"]

**Limitations:**

Yes

**Quality:**

3

**Strengths And Weaknesses:**

This paper presents a compelling approach to improving language modeling while maintaining a high degree of interpretability. Conceptually, the architecture draws a neat parallel with modern Transformers: the Infini-gram module resembles an interpretable feed-forward network, retrieving world knowledge from an external corpus, while the Generalized Induction-Head Model (GIM) plays a role analogous to attention, capturing local, in-context patterns in a transparent way. This line of work is important for probing the performance limits of interpretable models. Insights from this research could help guide the design of future architectures that are both effective and transparent.

That said, the current model still falls short of state-of-the-art black-box LLMs in terms of raw performance—despite notable gains over prior interpretable baselines. A promising direction for future work would be to explore hybrid approaches that integrate GIM-like mechanisms with black-box models, especially for tasks requiring strong in-context learning. Such combinations may strike a better balance between interpretability and capability. Overall, I see this as a valuable and timely contribution to the field.

---

> ### Author Rebuttal · Authors · 2025-07-30
>
> We thank the reviewer for their thoughtful and encouraging comments. We are especially grateful for the recognition of (1) our model’s interpretability and (2) its utility in exploring the performance boundaries of fully interpretable approaches. Below, we address each of the reviewer’s points in detail.
>
> ### W1. Raw performance gap vs. black-box LLMs & Hybrid directions
> We agree that GIM, while substantially improving over prior interpretable baselines, does not yet match the performance of state-of-the-art black-box LLMs, and that bridging this gap via hybrid approaches is a promising future direction. In fact, we have taken a first step in this direction using speculative decoding, as described in Appendix A.5. Specifically, we employ GIM as the draft model and a black-box LLM (LLaMA-2-7B/13B/70B) as the large model. This setup leverages the interpretability of GIM for generating candidate tokens while relying on the large model for final validation. Because GIM’s predictions are fully decomposable into exact or fuzzy n-gram matches, every accepted token from the draft stage carries transparent attribution, thus maintaining interpretability even when coupled with a powerful black-box verifier. We view such GIM–LLM hybrids as a promising direction for not only faster inference but also improved balance between transparency and performance. We will revise the Discussion section to further highlight these trade-offs and future directions (please see our response to Reviewer WRxr, **W2: GIM’s scope and relation to LLM internals**).
>
> ### Q1. GIM as a layer or attention head
> Thank you for the suggestion. While our current implementation keeps GIM external to maintain full decomposability and interpretability, we agree that incorporating GIM as an internal component (e.g., as an attention head, similar to [1]) could open up interesting trade-offs. Our current design emphasizes transparency by ensuring that all probability mass originates from explicit matches. Embedding GIM within model internals might dilute this property, but it may also unlock new capabilities. We mention this as a promising direction for future work (Lines 301-302), particularly if such integration can retain partial attribution while improving flexibility.
>
> We again thank the reviewer for their insightful and constructive feedback, and we hope our responses adequately address your concerns.

---

> > ### Comment · Reviewer_vuy2 · 2025-08-05
> >
> > Thank you for the rebuttal. I am keeping my positive score. While I appreciate the effort in interpretability research, its impact would be significantly greater if these methods could also improve black-box large language models. I therefore strongly recommend trying GIM as a layer or attention head.

---

> > > ### Author Response · Authors · 2025-08-06
> > >
> > > We thank the reviewer for their attentive review of our rebuttal and for the thoughtful comments. We agree that exploring ways to integrate GIM into black-box large models—such as using it as a layer or attention head—is an exciting and valuable direction. While our current work focuses on proposing GIM as a fully interpretable module, we appreciate the suggestion and will consider it as a promising avenue for future extensions.

---

### Note · Authors · 2025-08-15

We thank the reviewers for their careful reading of our paper, for providing constructive feedback, and for engaging with our rebuttal and subsequent discussions. In their initial reviews, they positively recognized our contributions in: (1) extending mechanistic interpretability by explicitly modeling the induction-head mechanism, (2) integrating Infini-gram with GIM to leverage context, (3) demonstrating transparency through n-gram-based reasoning, and (4) validating the model across both language and neuroscience domains.

A key concern raised in the initial reviews was the role and purpose of GIM. In our rebuttal, we explained that GIM is designed to maximize interpretability and to demonstrate how targeted mechanisms can be explicitly isolated, examined, and extended through interpretable design, rather than to replicate the full capabilities of LLMs, serve as a stand-alone replacement, or function as a diagnostic tool. This distinction is further reinforced by our comparison with attribution-based explanations for black-box LLMs. Reviewers acknowledged these clarifications, noting that they made the paper’s contributions much clearer.

In addition, we conducted constructive discussions supported by additional experiments and explanations, which helped address the remaining concerns from reviewers. We are grateful for the opportunity to incorporate this valuable feedback.

---

### Decision · Program_Chairs · 2025-09-17

**Decision:**

Accept (poster)

**Comment:**

The authors introduce the Generalized Induction-Head Model (GIM) as an "interpretable" model. In particular, GIM generates next tokens as a function of completions to matched phrases, permitting explicit attribution to matched phrases.

While there were some reservations from certain reviewers about framing the experimental setup (e.g., 3WB2), all reviewers nonetheless found this work compelling: It is a nice example of drawing on insights from mech interp to inform model design. The empirical results are compelling, and the work is likely to be of relatively wide interest to the NeurIPs community.